# Chemotherapy-induced complement signaling modulates immunosuppression and metastatic relapse in breast cancer

Lea Monteran[1,4], Nour Ershaid [1,4], Hila Doron[1], Yael Zait[1], Ye'ela Scharff[1], Shahar Ben-Yosef[1], Camila Avivi[2], Iris Barshack[2], Amir Sonnenblick [3] & Neta Erez [1] ✉

Mortality from breast cancer is almost exclusively a result of tumor metastasis and resistance to therapy and therefore understanding the underlying mechanisms is an urgent challenge. Chemotherapy, routinely used to treat breast cancer, induces extensive tissue damage, eliciting an inflammatory response that may hinder efficacy and promote metastatic relapse. Here we show that systemic treatment with doxorubicin, but not cisplatin, following resection of a triple-negative breast tumor induces the expression of complement factors in lung fibroblasts and modulates an immunosuppressive metastatic niche that supports lung metastasis. Complement signaling derived from cancer-associated fibroblasts (CAFs) mediates the recruitment of myeloid-derived suppressor cells (MDSCs) to the metastatic niche, thus promoting T cell dysfunction. Pharmacological targeting of complement signaling in combination with chemotherapy alleviates immune dysregulation and attenuates lung metastasis. Our findings suggest that combining cytotoxic treatment with blockade of complement signaling in triple-negative breast cancer patients may attenuate the adverse effects of chemotherapy, thus offering a promising approach for clinical use.

Mortality from breast cancer is almost exclusively a result of tumor metastasis. Lungs are one of the most common sites of breast cancer metastasis, conferring a median survival of less than 2 years after diagnosis, thus posing a major clinical challenge[1]. Triple-negative breast cancer (TNBC) is the most aggressive subtype of breast cancer[2]. TNBC is defined by the lack of expression of hormonal receptors and absence of HER2 overexpression, and thus its management relies mostly on chemotherapeutic agents. Anthracycline-taxane-based chemotherapy is currently the standard of care treatment option for most breast cancer types. However, improvement of TNBC treatment regimens is still an urgent need.

The main purpose of cytotoxic treatment following surgery (adjuvant treatment) is to control micrometastatic disease and reduce recurrence rate. Unfortunately, between 15 and 30% of women initially diagnosed with breast cancer will relapse with distant metastasis[3,4]. Accumulating evidence suggested that systemic changes induced by chemotherapy might paradoxically promote survival and growth of disseminated cancer cells at distant organs[5–8], thus abating the beneficial effects of chemotherapy. These pro-tumorigenic adverse effects are largely due to the induction of tissue damage and tumor-promoting inflammation.

The metastatic microenvironment is a crucial facilitator of seeding and growth of disseminated tumor cells[9]. Moreover, the crosstalk between stromal and immune cells at the metastatic niche contributes to the formation of an immunosuppressive microenvironment that modulates the therapeutic response of metastatic disease[6,10–12].

[1]Department of Pathology, Sackler Faculty of Medicine, Tel Aviv University, Tel Aviv, Israel. [2]Department of Pathology, Sheba Medical Center, Tel Hashomer, Tel Aviv, Israel. [3]Oncology Division, Tel Aviv Sourasky Medical Center, Sackler Faculty of Medicine, Tel Aviv University, Tel Aviv, Israel. [4]These authors contributed equally: Lea Monteran, Nour Ershaid. ✉e-mail: netaerez@tauex.tau.ac.il

Cancer-associated fibroblasts (CAFs) are a heterogeneous population of stromal cells shown to play a central role as mediators of tumor growth and metastasis[13–15]. Moreover, the crosstalk of CAFs with immune cells was shown to modulate tumor-associated immune responses[16–19]. We and others have previously demonstrated the functions of CAFs in modulating the immune microenvironment in mouse and human carcinomas[14,20–24]. Specifically, CAFs were implicated in including pro-inflammatory changes that foster a hospitable niche for breast cancer pulmonary metastasis[25–27]. However, very little is known about the role of fibroblasts as mediators of chemotherapy-induced host response at the metastatic niche.

The complement system is a central component of innate immunity comprising an extensive network of plasma and membrane-bound proteins, cofactors, and receptors. Inappropriate complement activation is associated with a variety of pathological conditions, including autoimmune disorders, neurodegenerative diseases, and cancer[28]. Complement activation induces the production of the highly potent anaphylatoxins C3a and C5a, which have profound effects on immunity[29] and on tumor-promoting inflammation[30,31]. Fibroblasts were recently implicated in complement production that enhances inflammation[32–34]. However, the role of CAFs and complement signaling in modulating response to chemotherapy is largely unknown.

In this study, we set out to characterize chemotherapy-induced alterations in the metastatic lung microenvironment. We find that adjuvant treatment with doxorubicin in two murine models of TNBC induces the formation of an immunosuppressive metastatic niche and is ineffective in hindering pulmonary metastasis. Systemic chemotherapy treatment following resection of the primary tumor upregulates the expression of complement components in lung fibroblasts, leading to the recruitment of myeloid-derived suppressor cells (MDSCs) that promote T cell dysfunction. Functionally, we show that pharmacological blockade of complement signaling in combination with chemotherapy reverses chemotherapy-induced immunosuppressive modulation and attenuates metastatic relapse, thus offering a promising approach for clinical use.

## Results

### Cytotoxic treatment alters the stromal and immune landscapes and enhances lung metastatic colonization

We previously demonstrated that CAFs function as sensors of tissue damage during breast cancer progression and metastasis[21]. We, therefore, hypothesized that fibroblasts may also function as sentinels of tissue damage caused by cytotoxic treatment.

To investigate the direct effect of chemotherapy-induced tissue damage on lung fibroblasts, we treated normal mice with systemic chemotherapy in a tumor free setting (Fig. 1a). We used doxorubicin, an anthracycline-based chemotherapy, commonly applied as first line chemotherapy of breast cancer[35], and cisplatin, a platinum-based chemotherapy, which is frequently incorporated into treatment regimens for TNBC and metastatic breast cancer[36]. Following treatment, we isolated lung fibroblasts (CD45⁻CD31⁻EpCAM⁻PDGFRα⁺) and analyzed changes in their gene expression. Analysis revealed that lung fibroblasts upregulated a pro-inflammatory gene signature upon systemic treatment with either doxorubicin or cisplatin (Fig. 1b). In addition to their role in promoting inflammation, ECM modulation is a central feature of CAFs. We, therefore, analyzed the expression of genes related to ECM modulation following systemic chemotherapy. Interestingly, while inflammatory genes were upregulated by both cytotoxic treatments, the expression of a fibrosis-related gene signature was downregulated in lung fibroblasts following treatment with doxorubicin (Fig. 1b). Furthermore, cell migration and contractility of lung fibroblasts were significantly decreased following cytotoxic treatment with both drugs (Supplementary Fig. 1a–d), suggesting that chemotherapy-induced tissue damage promotes functional differentiation of lung fibroblasts towards a pro-inflammatory phenotype.

Since many of the genes that were upregulated in lung fibroblasts were potent chemoattractants of myeloid cells such as CXCL1, CXCL2, CXCL5 and CCL2, we next assessed the effect of chemotherapy-induced host response on changes in the lung immune milieu (Fig. 1c). Analysis of the results indicated remodeling of the lung immune microenvironment of mice following cytotoxic treatment (Supplementary Fig. 1e). Specifically, analysis revealed a significant increase in granulocytes (CD11b⁺Ly6CⁱⁿᵗLy6G⁺) and eosinophils (CD11b⁺CD11c⁻Ly6G⁻SiglecF⁺) following administration of either doxorubicin or cisplatin (Fig. 1d, e), while T cells (CD3⁺) were only increased in lungs of cisplatin-treated mice (Fig. 1f). Interestingly, both drugs led to a decrease in the monocytic population (CD11b⁺Ly6C⁺Ly6G⁻) (Fig. 1g).

We next asked whether this host response to systemic chemotherapy affects breast cancer cell seeding and growth in lungs. To that end, mice were treated with systemic chemotherapy, followed by intravenous injection with TNBC cells expressing a luciferase reporter gene (4T1-luc) (Fig. 1h). Markedly, cancer cell colonization in mice treated with doxorubicin or with cisplatin was significantly enhanced compared to control mice (Fig. 1i–k). These findings suggest that systemic chemotherapy treatment directly induces changes in the lung microenvironment, even in the absence of a primary tumor, which may paradoxically facilitate metastatic colonization of breast cancer cells by creating a hospitable niche.

### Adjuvant doxorubicin treatment is ineffective in curbing lung metastasis and drives an immunosuppressive niche in lungs

Intrigued by these findings, we set out to investigate chemotherapy-induced host response in a clinically relevant setting of spontaneous pulmonary metastases, following surgical resection of a primary TNBC tumor. Mice were orthotopically injected with 4T1 breast cancer cells. Following tumor resection, mice were treated with two cycles of doxorubicin or cisplatin, recapitulating the standard of care for breast cancer patients (Fig. 2a). Lung metastatic load was assessed intravitally by CT imaging, followed by H&E staining of lung tissue sections (Fig. 2b). Surprisingly, analysis of the results revealed that doxorubicin was ineffective in reducing metastatic load or incidence, while cisplatin at the same setting significantly hindered both the metastatic load and incidence compared to either control or doxorubicin (Fig. 2c–e). Of note, doxorubicin and cisplatin were equally effective in killing 4T1 cancer cells in vitro (Supplementary Fig. 1f, g), implying that differences in host responses to distinct cytotoxic agents might affect the overall reaction to chemotherapeutic treatment and determine the metastatic outcome. To ensure that our findings are not model-specific, we performed similar experiments in an additional mouse model of transplantable TNBC, using the EO771 cell line[37]. Mice were treated with chemotherapy in adjuvant setting, following primary tumor resection (Fig. 2f). Interestingly, analysis of lung metastasis revealed that similarly to the 4T1 model, doxorubicin was inefficient in attenuating pulmonary metastasis as compared to cisplatin therapy (Fig. 2g, h). These findings prompted us to further study the mechanisms underlying the paradoxical effects of doxorubicin therapy on host microenvironment that potentially promote chemoresistance.

Modifications of the lung metastatic niche were shown to precede the formation of metastases[27,38,39]. We, therefore, hypothesized that the observed discrepancy in metastatic recurrence in mice treated with doxorubicin compared with cisplatin and control mice is associated with alterations in the immune microenvironment. Utilizing the adjuvant treatment experimental platform, we analyzed lungs of mice 24 h following the final dose of chemotherapy (Fig. 3a). Strikingly, while almost none of the cisplatin- or vehicle-treated mice presented with lung macro-metastases at this early metastatic stage (6 and 7% respectively), doxorubicin-treated mice developed significantly more early macro-metastases (32%, Fig. 3b). These results further suggest that doxorubicin might have adverse effects on lung metastasis. However, at later stages, the metastatic load of dox-treated mice was

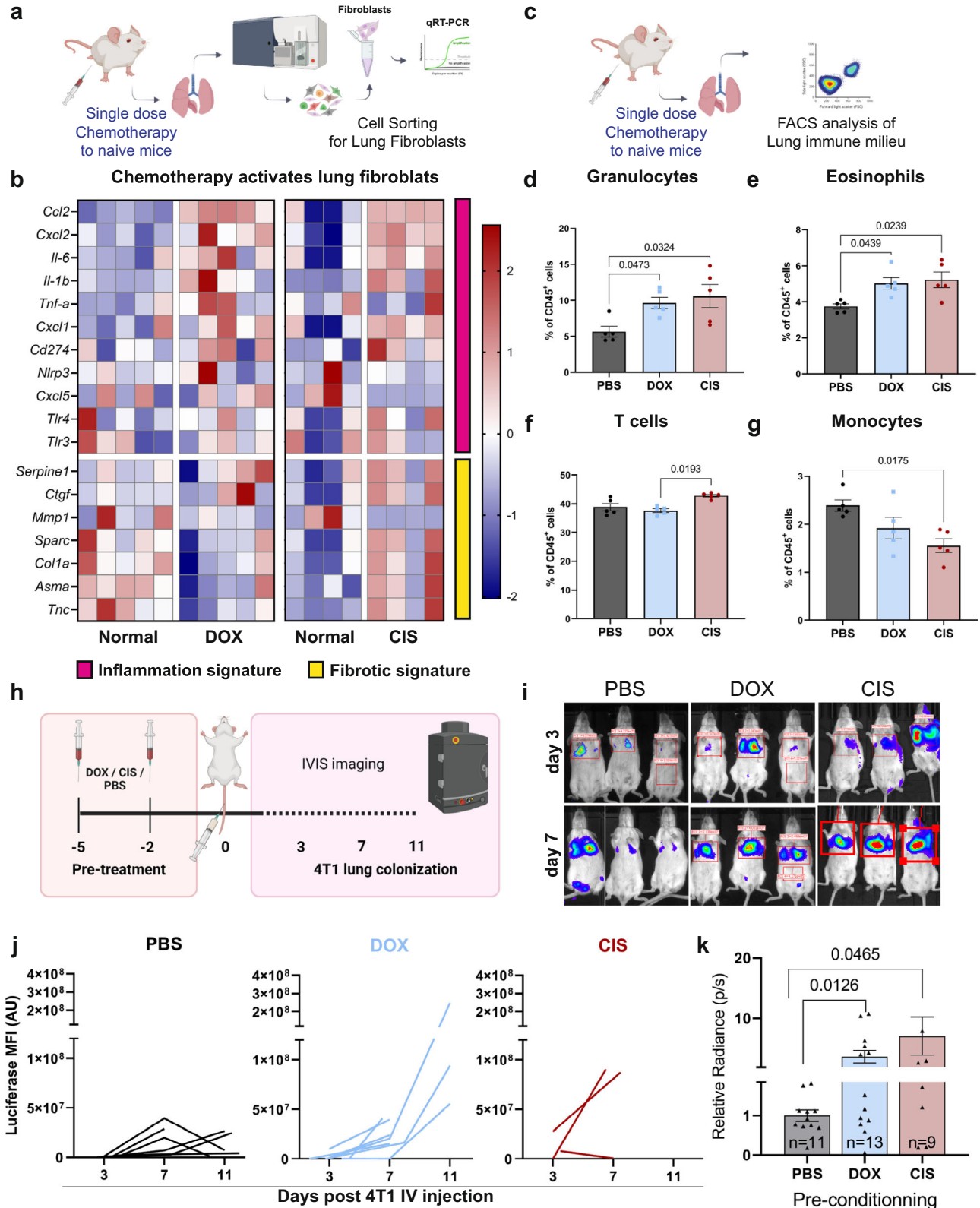

**b** **Chemotherapy activates lung fibroblats**

Inflammation signature · Fibrotic signature

**d** Granulocytes **e** Eosinophils **f** T cells **g** Monocytes

**h** Pre-treatment / 4T1 lung colonization

**i** PBS DOX CIS — day 3 / day 7

**j** PBS DOX CIS — Luciferase MFI (AU) / Days post 4T1 IV injection

**k** Relative Radiance (p/s) / Pre-conditionning

comparable to no treatment (Fig. 2e, h). This discrepancy may be explained by the fact that the overall effect of chemotherapy is the sum of its beneficial effects (killing of cancer cells) and its deleterious adverse effects, including the induction of tissue damage and inflammation. Our findings suggest that at early stages, the tissue damage and tumor-promoting inflammation induced by doxorubicin counteract the anti-cancer effects and allow for more metastases formation

compared with control, while at later stages, the balance of these two contradicting effects resulted in no therapeutic benefit.

We next analyzed the immune composition of lungs (Supplementary Fig. 2a, b) following chemotherapy treatment at early metastatic stage in both mouse models: 4T1 (Fig. 3c–f; Supplementary Fig. 2c–i) and EO771 (Fig. 3d–f; Supplementary Fig. 3). We found significant differences in the lung immune composition between the

**Fig. 1 | Cytotoxic treatments induce alterations in the stromal and immune landscapes and enhance breast cancer cell colonization in lungs. a** Gene expression analysis of sorted lung fibroblast (PDGFRα⁺) following chemotherapy. One dose of doxorubicin (DOX, 5 mg/kg), cisplatin (CIS, 5 mg/kg), or saline (PBS) was administered to naive BALB/c mice by intraperitoneal (i.p) injection. $n = 5$ (Dox), $n = 4$ (CIS). **b** Heatmap of inflammatory and fibrosis-related gene expression. **c** Flow cytometry analysis of major immune cell populations at lungs following one dose treatment with chemotherapy. Flow cytometry analysis of: **d** granulocytes (Ly6G⁺Ly6C^int), **e** eosinophils (Ly6G⁻SiglecF⁺), **f** T cells (CD3⁺), **g** monocytes (Ly6G⁻Ly6C⁺). $n = 5$ mice per group. Data presented are mean percentage of CD45⁺ cells ± s.e.m; *P*-values were calculated using one-way ANOVA test. **h** Lung

colonization assay. Naive BALB/c mice were pre-treated with two injections of chemotherapy or PBS as control, and 48 h later 2*10⁵ 4T1-*luciferase* cells (4T1-luc) were injected intravenously (IV) and early lung colonization was monitored using IVIS imaging. **i** Representative IVIS imaging, 3 and 7 days post-inoculation. **j** Kinetics of 4T1 lung metastatic growth represented by luciferase bioluminescence intensity following 4T1-luc IV injection. **k** Quantitative analysis of data shown in (**i, j**) at day 7. $n = 11, 13$, and 9 in PBS, DOX, and CIS groups, respectively. Data from two independent experiments are presented as mean ± s.e.m, normalized to PBS group; *P*-values were calculated using one-tailed Welch's t-test. Graphical summary was designed using BioRender. Source data are provided as a Source Data file.

treatments. Specifically, in both models, lungs of doxorubicin-treated mice were highly infiltrated by granulocytes and monocytes, compared with cisplatin-treated mice or control mice (Fig. 3d, e). Of note, lung metastasis is associated with disequilibrium of the immune microenvironment even without cytotoxic treatment (Supplementary Fig. 4a–j). However, doxorubicin treatment but not cisplatin further amplified the infiltration of granulocytes and monocytes.

Analysis of T cell infiltration revealed that treatment with cisplatin, but not with doxorubicin enhanced T cell recruitment to lungs in the 4T1 model (Fig. 3f). T cell recruitment is often associated with favorable prognosis[40], in agreement with our findings that cisplatin had an inhibitory effect on lung metastasis (Fig. 2b–h).

Interestingly, the effect of chemotherapy in lungs was also evident in the spleen of doxorubicin-treated mice, which exhibited increased infiltration of granulocytes and monocytes (Fig. 3g and Supplementary Fig. 4k–m), consistent with its role as a major reservoir of myeloid cells in cancer[41]. However, the changes in the lung immune milieu following cytotoxic treatment were not evident in peripheral blood, suggesting that these events are lung-specific (Fig. 3h, Supplementary Fig. 4n–p).

Since tissue damage and inflammation are often accompanied by ECM modifications[42], we also analyzed collagen deposition in lungs of treated mice by Sirius Red staining. Interestingly, we found significantly elevated collagen deposition in lungs of mice treated with doxorubicin but not with cisplatin (Supplementary Fig. 4q–s). These findings are in agreement with recent studies showing a role for increased collagen deposition in mediating chemotherapy-induced pulmonary metastasis[43].

In the clinic, doxorubicin treatment for TNBC is usually included in the "AC regimen" in combination with cyclophosphamide (CTX). To assess if the "AC regimen" induces similar adverse effects on host microenvironment, we treated mice with a combination of DOX + CTX following primary tumor resection (Supplementary Fig. 5a,h). Analysis of the immune milieu in treated mice revealed an increase in infiltration of granulocytes and monocytes into lungs, accompanied by increased T cells and decreased NK and B cells (Supplementary Fig. 5a–g). However, while some of these changes were significantly different from those induced by doxorubicin alone, there was no additional benefit of the AC regimen on metastatic load at end-stage, compared with doxorubicin alone, or with no treatment (Supplementary Fig 5h–j).

To get further insight on the mechanisms underlying the diminished chemotherapy efficacy, we focused our further analyses on the early changes in the lung metastatic niche following adjuvant doxorubicin treatment. Since doxorubicin enhanced recruitment of myeloid cells into lungs, we isolated infiltrated granulocytes and monocytes from lungs of doxorubicin-treated mice or control mice following primary tumor resection and analyzed their gene expression 24 h after the last treatment dose (Fig. 4a). The results indicated that both myeloid cell populations isolated from doxorubicin-treated mice upregulated the expression of an immunosuppression-related gene signature, typical of MDSCs (Fig. 4b, c). Upregulated genes included known MDSC markers such as PD1-L1, IL-6, IL-1β, IL-4Ra, and IL-10[44,45]. Analysis of a combined immunosuppressive score (comprised of genes

depicted in heatmaps in Fig. 4b, c) further confirmed that doxorubicin treatment instigated the expression of MDSC gene signature in recruited myeloid cells (Fig. 4d). Interestingly, analysis of the immunosuppressive gene expression in cisplatin-treated mice, revealed a stark difference compared with doxorubicin-treated mice: while the expression of immunosuppressive genes in lung immune cells (CD45⁺ cells) was upregulated by doxorubicin, cisplatin-treated mice presented reduced expression of those genes when compared to control-mice, providing further mechanistic insights on the differences between doxorubicin and cisplatin (Supplementary Fig. 6a–c).

One of the pathways by which MDSCs exert their pro-tumorigenic effects is by restricting T cell function[44,45]. Therefore, we assessed the expression of dysfunction markers on lung infiltrating T cells. Importantly, lungs of doxorubicin-treated mice had increased infiltration of T cells expressing exhaustion/dysfunction markers such as TIM-3, TIGIT, and PD-1 (Fig. 4e, f and Supplementary Fig. 6d). To functionally validate these findings, we analyzed the suppressive activity of MDSCs isolated from spleens of doxorubicin-treated mice or from control mice on T cell proliferation (Fig. 4g). Notably, MDSCs derived from doxorubicin-treated mice had greater capacity to inhibit T cell proliferation compared with controls (Fig. 4h). Thus, chemotherapy induces an immunosuppressive metastatic niche following systemic treatment, at early metastatic stages.

Considering that T cell dysfunction is exacerbated at advanced stages of tumorigenesis, we assessed whether doxorubicin-induced modulation of T cell functional state was maintained at later metastatic stages by analyzing T cells isolated from lungs of treated mice at end stage. As expected, we observed higher proportion of cytotoxic T cells expressing dysfunction markers in both control and doxorubicin-treated mice at end-stage (Fig. 4i, j), compared with T cells isolated from mice during early metastatic disease (Fig. 4e, f). Moreover, in agreement with our results in early metastatic stages, T cells in lungs of doxorubicin-treated mice with advanced metastasis displayed upregulated expression of dysfunction markers compared with control treated mice (Fig. 4i, j). This was also evident in EO771-injected mice at late metastatic stage (Supplementary Fig. 6e–g).

Taken together, these findings suggest that adjuvant treatment with doxorubicin promotes increased recruitment of MDSCs and enhances the formation of an immunosuppressive microenvironment in the lung metastatic niche.

## Complement system signaling is upregulated in lung fibroblasts following doxorubicin treatment

Since our initial observations indicated that fibroblasts respond to systemic chemotherapy by instigating pro-inflammatory signaling (Fig. 1), we hypothesized that some of the immune modifications observed following adjuvant chemotherapy are mediated by reprogrammed fibroblasts. To test this, we isolated fibroblasts in the lung metastatic niche, utilizing the resectable TNBC model followed by adjuvant doxorubicin treatment. Fibroblasts were isolated from lungs of mice treated with doxorubicin or control mice 24 h following the last treatment dose, and their inflammatory transcriptome was analyzed (Fig. 5a and Supplementary Fig. 6h). Indeed, we found that fibroblasts

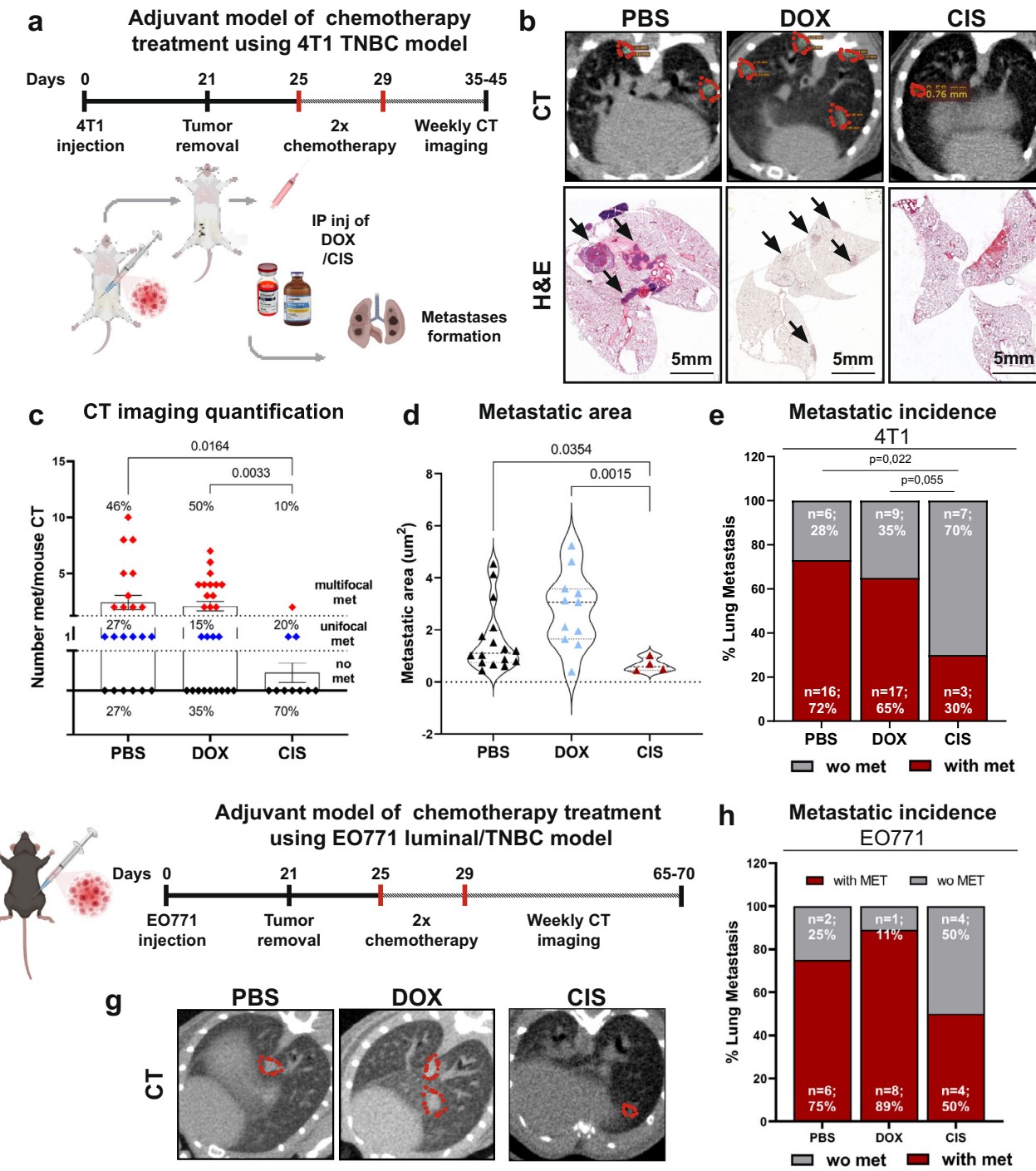

**Fig. 2 | Adjuvant doxorubicin is ineffective in restricting lung metastasis of breast cancer. a** Murine model of TNBC combined with adjuvant chemotherapy. 4T1 cells were orthotopically injected into the mammary fat pad of BALB/c mice. 3 weeks post-inoculation primary tumors were resected, and mice received two cycles of adjuvant doxorubicin (5 mg/kg), cisplatin (5 mg/kg), or PBS as control by i.p. injection. Spontaneous lung metastases were monitored intravitally by CT imaging. **b** Representative images of CT-scans and H&E staining of lungs at late-metastatic stages. Metastases are circled in red or indicated by black arrows in CT scans and H&E slides, respectively. Scale bars, 5 mm. **c** Quantification of number of lung metastatic foci identified by CT imaging. Data presented as mean ± s.e.m. Percentage refers to distribution among subgroups (multifocal met, unifocal met, or no met); *P*-values were calculated using Brown–Forsythe ANOVA multiple comparison test. *n* = 22, 26, and 10 mice in control, doxorubicin, and cisplatin,

respectively. **d** Quantification of total metastatic area in lung CT images. Data presented as mean; *P*-values were calculated using one-way ANOVA test. *n* = 9 mice per group were analyzed, graph presents metastatic areas from metastatic bearing mice only. **e** Metastatic incidence in the 4T1 model, analyzed by CT imaging and H&E staining. P-values were calculated using two-sided Chi square test. *n* = 22, 26, and 10 mice in control, doxorubicin, and cisplatin, respectively. **f** EO771 model of TNBC combined with adjuvant chemotherapy. EO771 cells were orthotopically injected into the mammary fat pad of C57BL/6 mice. Primary tumors were resected, and mice received adjuvant chemotherapy as described above. Spontaneous lung metastases were monitored intravitally by CT imaging. **g** Representative images of CT scans. **h** Metastatic incidence in the EO771 model. *n* = 8, 9, and 8 mice in control, doxorubicin, and cisplatin, respectively. Graphical summary was designed using BioRender. Source data are provided as a Source Data file.

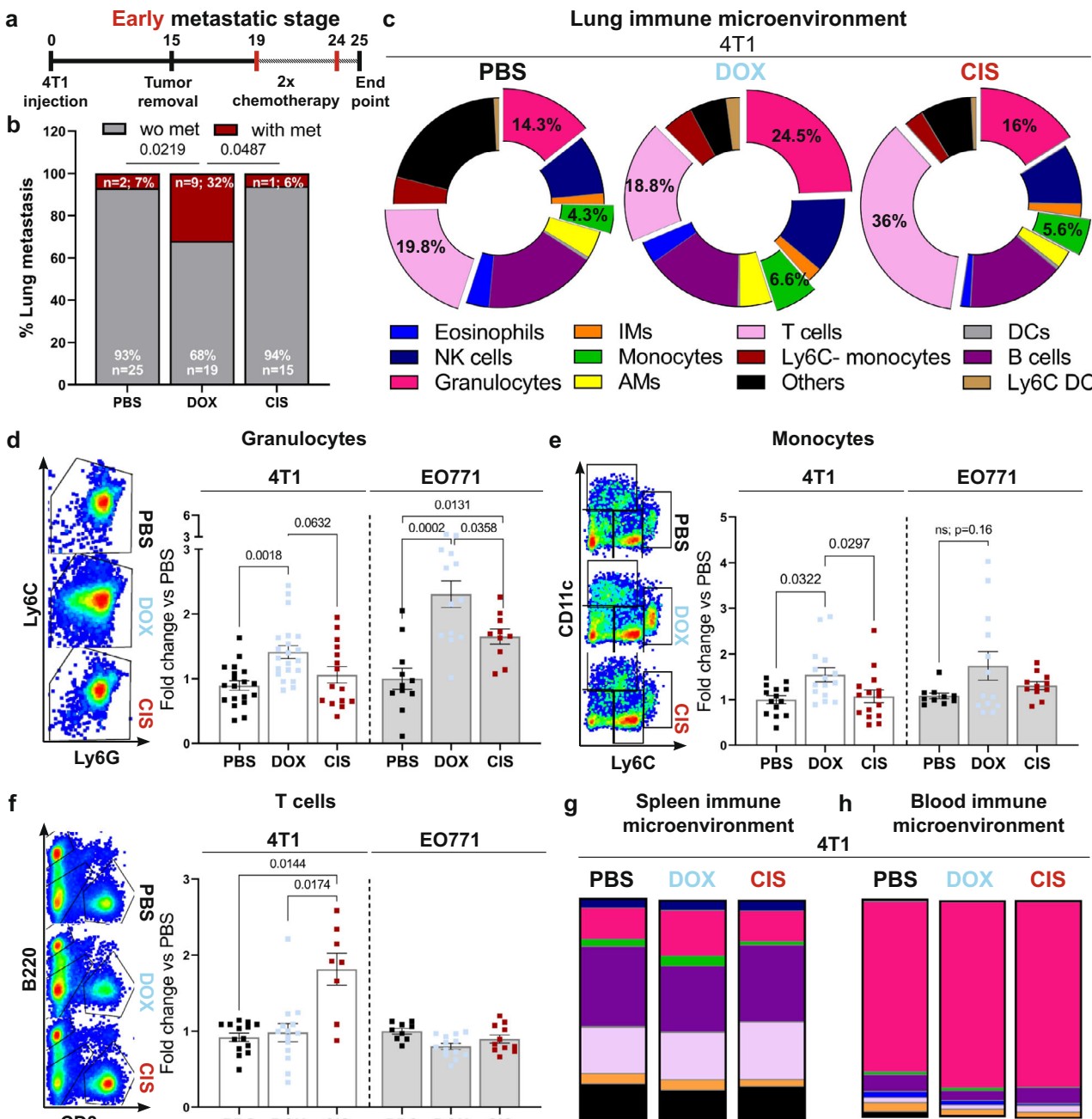

**Fig. 3 | Doxorubicin treatment disrupts the immune milieu in the lung metastatic niche. a** Adjuvant treatment in early-stage metastatic breast cancer models used in (**b**–**h**). Following tumor removal (2w post-inoculation), mice received two cycles of chemotherapy, or PBS as control. **b** Incidence of lung macro-metastases at early stages; *n* = 27, 28, and 16 mice in control, doxorubicin, and cisplatin, respectively. *P*-values were calculated using two-sided Chi square test. **c**–**f** Immune landscape of lungs at early metastatic stages, analyzed by flow cytometry. **c** Proportions of various immune cell populations in lungs. Representative plots and quantification of most-altered populations from 4T1 and EO771 models:

**d** Granulocytes; **e** Monocytes, and **f** T cells. *n* = 21, 15, and 19 mice in control, doxorubicin, and cisplatin, respectively in the 4T1 model, and *n* = 11, 13 and 10 mice in control, doxorubicin, and cisplatin, respectively in the EO771 model. Data present three independent experiments. Immune landscapes of spleen and blood. Colors in (**g**) and (**h**) depict immune cell populations labeled as in (**c**). In **d**–**f**, data are presented as mean percentage out of CD45⁺ cells, normalized to PBS group from each model. Error bars represent s.e.m; P-values were calculated using Kruskal–Wallis test for multiple comparisons. Source data are provided as a Source Data file.

isolated from lungs of mice treated with doxorubicin were transcriptionally reprogrammed compared to control mice (Supplementary Fig. 6i, j), suggesting that lung fibroblasts may play a functional role in chemotherapy-induced host response. Surprisingly, further analysis of the results indicated upregulation in the expression of multiple genes from the complement system in lung fibroblasts of doxorubicin-treated mice, including C1ra, C1s, C2, C3, C4a, C5, C7, C8a, C8b (Fig. 5b). GSEA analysis confirmed enrichment of the complement cascade in

fibroblasts from doxorubicin-treated mice (Fig. 5c). Interestingly, C3a and C5a were previously shown to promote tumor progression and metastasis by skewing T cell mediated anti-tumor immunity[46–48], in part due to enhanced deployment and activation of MDSCs[49,50]. Intrigued by these findings, we hypothesized that fibroblast-derived complement signaling may mediate the enhanced recruitment of MDSCs to lungs of doxorubicin-treated mice, consequently favoring the formation of an immunosuppressive microenvironment.

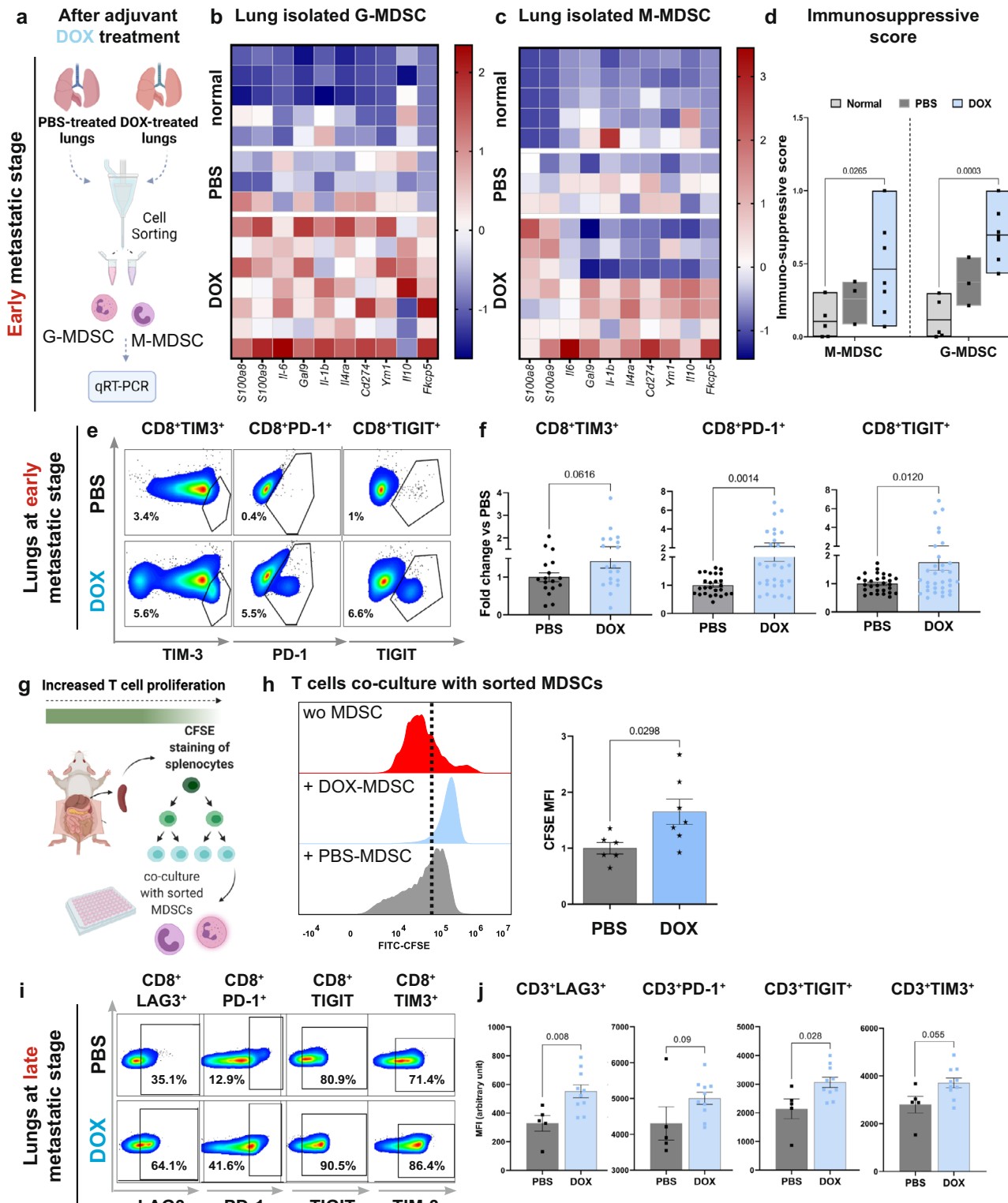

To further validate the cellular origins of complement factors in the lung metastatic microenvironment, we isolated by flow cytometry fibroblasts (CD45⁻CD31⁻EpCam⁻PDGFRα⁺), immune cells (CD31⁻EpCam⁻PDGFRα⁻CD45⁺), endothelial cells (EpCam⁻PDGFRα⁻CD45⁻CD31⁺), and epithelial cells (CD31⁻PDGFRα⁻CD45⁻EpCam⁺) from lungs of mice at early metastatic stages (Fig. 5a and Supplementary Fig. 6h). Analysis of the expression of complement components indicated that fibroblasts are the main source of C2, C3, and C7 while C5 was also expressed by epithelial and endothelial cells (Fig. 5d). We next asked whether

expression of complement factors in lung CAFs is a general phenomenon. Analysis of a single-cell RNA-seq dataset of the stromal compartment from murine breast cancer lung metastasis confirmed our observations that lung CAFs are a major source of complement signaling (Supplementary Fig. 7a). Notably, analysis of two additional datasets of human lung tumors revealed that lung fibroblasts are also a major source of complement components in the tumor microenvironment of lung cancer patients (Fig. 5e and Supplementary Fig. 7b). Moreover, we analyzed a dataset of human breast cancer metastases from different

**Fig. 4 | Doxorubicin treatment drives an immunosuppressive microenvironment in lungs. a** Isolation of G-MDSCs and M-MDSCs from lungs of vehicle-treated mice (PBS-treated) or doxorubicin-treated mice (DOX-treated). **b, c** Heatmap of immunosuppression-related gene expression of G-MDSCs and M-MDSCs sorted from lungs at early metastatic stage. **d** Expression of immunosuppression-related genes were combined into an "immunosuppressive score". Data is presented as min to max, line represents mean. *P*-values were calculated using two-way ANOVA test. **b**–**d** $n$ = 7, 3, and 5 mice in doxorubicin, control, and normal, respectively. **e, f** Flow cytometry analysis of dysfunction markers in T cells isolated from lungs, 24 h following last treatment dose. **e** Representative flow cytometry plots of CD3$^+$CD8$^+$ T cells expressing dysfunction markers. **f** Quantification of changes in CD8$^+$ T cells expressing dysfunction markers. $n$ = 17 and 19 mice in PBS and doxorubicin, respectively. Data presented as mean percentage out of CD3$^+$ cells, normalized to

PBS group; *P*-values were calculated using Mann–Whitney test. **g** CFSE-based T cell proliferation assay following co-culture with MDSCs isolated from spleens of mice at early metastatic stages, 24 h following last treatment dose. **h** Representative histogram plot and quantification of CFSE intensity in stimulated splenocytes co-cultured with MDSCs ($n$ = 6 and 7 mice in PBS and DOX, respectively). Splenocytes cultured w/o MDSCs served as controls. **i, j** Flow cytometry analysis of dysfunction markers expression in lung-derived CD3$^+$CD8$^+$ cytotoxic T cells at late metastatic stages. **i** Representative flow cytometry plots of cytotoxic T cells expressing dysfunction markers. **j** Quantification of mean fluorescent intensity (MFI) of specified dysfunction markers in cytotoxic T cells. $n$ = 5, 10 mice in PBS and doxorubicin, respectively. Data presented as mean ± s.e.m.; *P*-values were calculated using Mann–Whitney test. Graphical summary was designed using BioRender. Source data are provided as a Source Data file.

organs and found that C2, C3, and C7 were significantly upregulated in lung metastases compared with brain or bone metastases, suggesting a specific role for complement signaling in pulmonary metastasis (Supplementary Fig. 7c).

To identify the signals that instigate the expression of complement factors in lung fibroblasts following doxorubicin treatment, we cultured primary lung fibroblasts with either 4T1-conditioned medium, doxorubicin alone, or conditioned medium (CM) of doxorubicin-treated 4T1, presumably containing damage-associated molecular patterns (DAMPs) released during chemotherapy-induced cell death (Fig. 5f). We found that C1ra, C2, and C3 were upregulated in lung fibroblasts incubated with CM of chemotherapy-treated cancer cells, while expression of C5 and CFP was directly upregulated upon incubation with doxorubicin (Fig. 5g–l). Moreover, the expression of complement factors was not upregulated in lung fibroblasts following exposure to cancer cell CM alone, indicating that although tumor-secreted factors upregulated the expression of known inflammatory mediators in lung fibroblasts (IL-1β, IL-6) (Supplementary Fig. 8a), they were not sufficient to trigger activation of complement signaling, at least in vitro. Interestingly, this upregulation may be specific to doxorubicin-induced damage response, as the expression of complement genes was not altered in lung fibroblasts incubated with CM from cisplatin-treated cancer cells (Supplementary Fig. 8b, c). Thus, activation of the complement cascade in lung fibroblasts is induced by direct effect of specific cytotoxic therapies, or as a response to specific damage-associated signals released from cancer cells.

## Activation of complement signaling in the lung metastatic niche is associated with infiltration of C3aR and C5aR1 expressing MDSCs

C3 is a central inflammatory mediator, shown to be important in several cancer types[30]. In the complement cascade, C3 is the precursor of both C3a and C5a, potent chemoattractants of immune cells. We, therefore, focused our further analyses on these factors. We confirmed that the levels of C3 deposits were elevated in lung tissue sections from doxorubicin-treated mice as compared to vehicle-treated mice already at early metastatic stage (Fig. 6a, b). Moreover, analysis of secreted C3a in lung supernatant (Fig. 6c) revealed that lungs of doxorubicin-treated mice exhibited higher levels of C3a following doxorubicin administration in both TNBC models (4T1, Fig. 6d and EO771 Supplementary Fig. 8d, e), which remained elevated at late metastatic stages (Fig. 6e). To validate the relevance of our findings to human disease, we analyzed the expression of C3 in a cohort of breast cancer patients with lung metastasis. Importantly, analysis of the results confirmed stromal deposition of C3 in the lung metastatic microenvironment (Fig. 6f–h), suggesting that stromal complement signaling is operative in human lung metastasis.

We further analyzed the expression of several complement components (C2, C3, C7, CFP, C3aR1, and C5aR1) and incorporated the results into a combined "complement score". Analysis indicated upregulation in the expression of the complement score in lungs of

mice at early metastatic stages following doxorubicin treatment in both 4T1- and EO771 models (Fig. 6i and Supplementary Fig. 8f). Moreover, the complement score positively correlated with increased lung micrometastases (Fig. 6j), as well as with an "immunosuppression score" (Fig. 6k). Notably, no augmentation of complement components expression or of C3a deposition in lungs was observed following cisplatin treatment, confirming that enhanced complement signaling in lungs is doxorubicin-specific (Supplementary Fig. 8d–h).

Seeking to further investigate the function of CAF-derived complement, we hypothesized that CAF-derived C3a and C5a, known to be chemoattractants, facilitate immune cell recruitment. We, therefore, analyzed the expression of the C3a and C5a receptors in distinct cell populations isolated from lungs of mice at early metastatic stages. We found that the anaphylatoxin receptors C3aR and C5aR1 were primarily expressed by immune cells in metastatic lungs (Fig. 7a). Analysis of scRNA-seq data of human lung tumors confirmed that this was the case also in human patients (Fig. 5e). Specifically, while C5aR1 was exclusively expressed in granulocytes, C3aR expression was evident in both monocytic- and granulocytic-MDSCs (Fig. 7b, c). Moreover, expression levels of C3 in lungs positively correlated with levels of both C3aR ($r$ = 0.7156, $p$ < 0.0004) and C5aR1 ($r$ = 0.6026, $p$ < 0.004) (Fig. 7d), suggesting that increased production of C3 in the lung metastatic niche results in enhanced recruitment of C3aR/C5aR1 expressing cells. Thus, CAF-derived complement signaling following cytotoxic treatment may contribute to the formation of an immunosuppressive metastatic niche.

## Combining chemotherapy with blockade of complement signaling attenuates lung metastatic relapse

To assess the functional significance of complement signaling to chemotherapy-induced inflammation, we tested whether pharmacological targeting of the C3a-C3aR and C5a-C5aR1 axes would alleviate the adverse effects of doxorubicin treatment on lung metastasis. We combined adjuvant chemotherapy treatment with complement receptor antagonists (SB290157 blocking C3aR, or PMX53 blocking C5aR1) following primary tumor resection (Fig. 7e). Analysis at early metastatic stage revealed normalization of lung "complement score" following combined doxorubicin and C3aR antagonist treatment, but no significant change was observed in the group treated with the combined doxorubicin+C5aR1 antagonist or in the groups receiving antagonists only (Fig. 7f and Supplementary Fig. 9a). Importantly, treatment with the C3aR antagonist reduced the recruitment of immune cells to the lung metastatic niche when combined with doxorubicin (Fig. 7g). We therefore performed a detailed analysis of the infiltrated immune cell populations in lungs of mice following the different treatments (Supplementary Fig. 9b). Analysis revealed that blocking C3a signaling in combination with doxorubicin treatment significantly attenuated the recruitment of G-MDSCs and eosinophils (Fig. 7h, i), and diminished the recruitment of M-MDSCs (which did not reach statistical significance, Fig. 7j). Inhibition of C5aR in combination with doxorubicin did not affect the recruitment of these myeloid cell

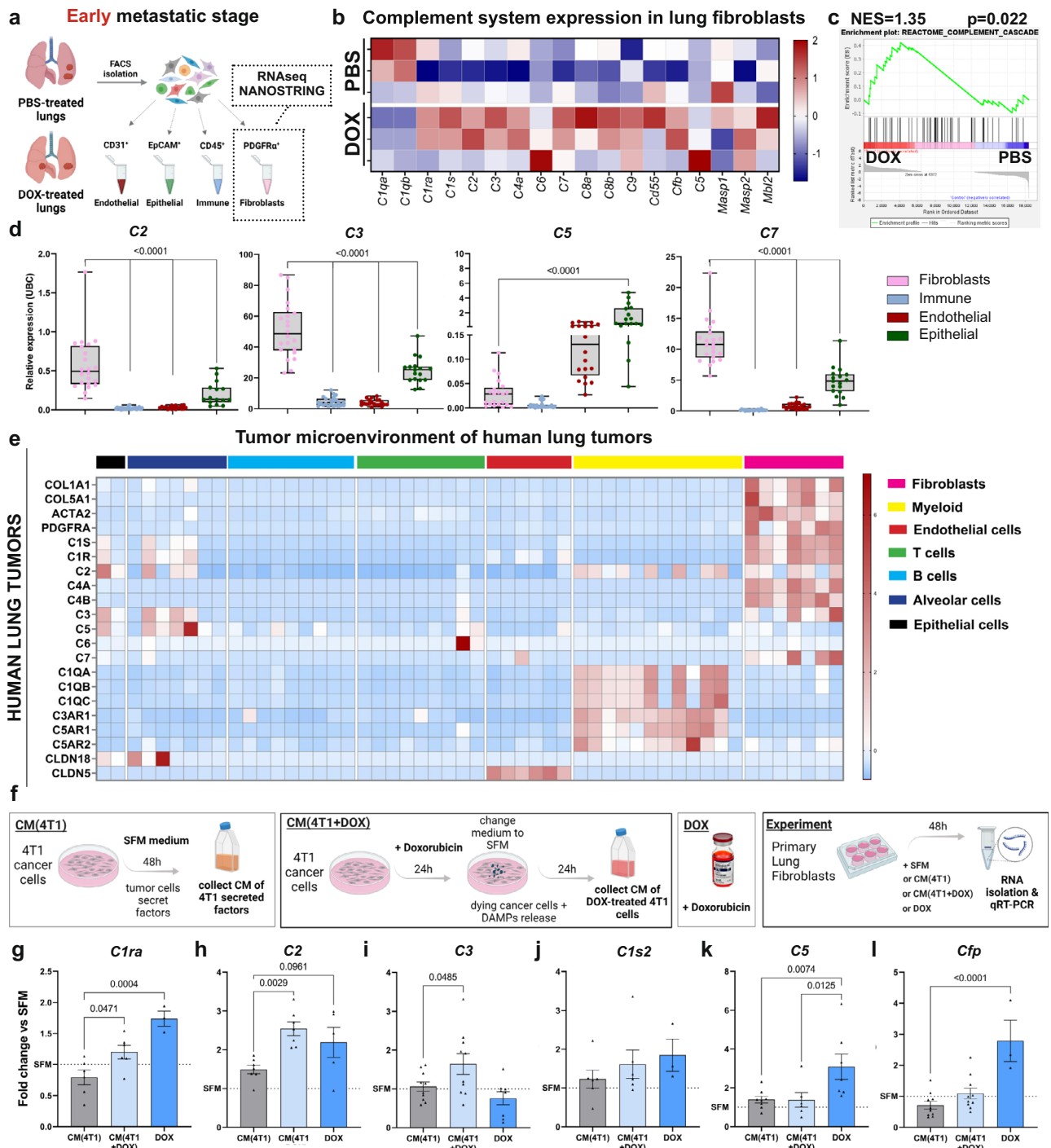

**Fig. 5 | Lung fibroblasts upregulate the expression of complement system factors following doxorubicin treatment. a** FACS sorting strategy for specified cell populations in early-stage metastatic lungs: transcriptome of sorted fibroblasts was analyzed using bulk-RNAseq or the NanoString nCounter Platform. **b** Heatmap of expression data from NanoString nCounter profiling. $n = 3$ mice per group. **c** Gene Set Enrichment Analysis (GSEA) of the gene set "complement cascade" in lung fibroblasts isolated from doxorubicin treated mice (DOX) compared with lungs of vehicle treated mice (PBS), $p < 0.022$, normalized enrichment score (NES = 1.35). $n = 4$ and 5 mice for PBS and DOX, respectively. **d** qRT-PCR analysis of complement components (*c2*, *c3*, *c5,* and *c7*) expression in cell populations isolated from lungs. $n = 23; 21; 18$ and 15 for CD45[+], PDGFRα[+], CD31[+] and EpCAM[+] respectively. Data are presented as relative expression to housekeeping gene (*ubc*); bars represent range of data points, line represents mean. *P*-values were calculated using one-way ANOVA followed by Tukey multiple comparisons test were performed. **e** Heatmap depicting expression of selected cluster-specific genes and complement-related genes derived from single-cell RNAseq data of human lung tumors: 52,698-cells from 5 patients were analyzed in this study[78]. **f** Experimental scheme for testing the direct and indirect effects of doxorubicin treatment on lung fibroblasts. **g–l** qRT-PCR analysis of complement factors in lung fibroblasts treated as described in (**f**). **g** $n = 6, 6$ and 3 **h** $n = 7, 7$ and 5 **i** $n = 10, 10$ and 8 **h** $n = 7, 7$ and 5 **j** $n = 6, 6$ and 3 **k** $n = 8, 6$ and 7 **l** $n = 10, 10$ and 3 biological repeats for CM(4T1), CM(4T1 + DOX) and DOX, respectively. Data are presented as mean ± s.d, normalized to SFM, and are representative of four biological repeats. *P*-values were calculated using one-way ANOVA. Graphical summary was designed using BioRender. Source data are provided as a Source Data file.

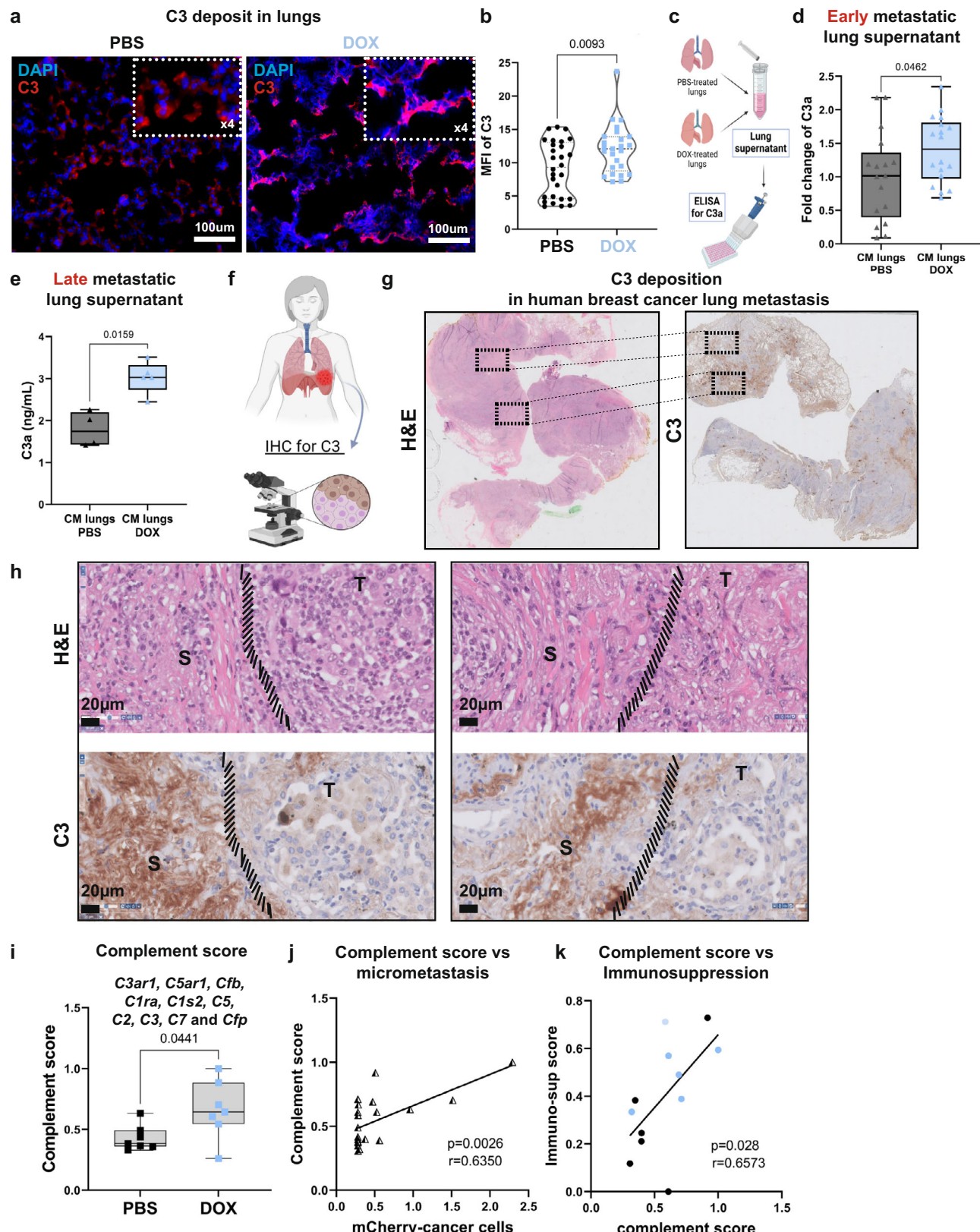

populations (Fig. 7h–j). Importantly, treatment with either one of the complement receptor antagonists in combination with doxorubicin resulted in a striking decrease of PD-1⁺ dysfunctional T cells in early metastatic lungs (Fig. 7k). However, treatment with complement receptor antagonists alone did not induce these effects (Supplementary Fig. 9b–g). Thus, combining chemotherapy treatment with

complement receptor blockade attenuated both the upregulation of complement signaling, and the immune dysregulation caused by doxorubicin treatment.

Finally, we tested whether blockade of the C3-C3aR and C5-C5aR1 axes would affect lung metastatic relapse. Mice were treated with adjuvant chemotherapy combined with anaphylatoxin receptor

**Fig. 6 | Activation of complement signaling in lungs is associated with infiltration of C3aR and C5aR1 expressing MDSCs. a** Representative images of C3 staining in lungs of doxorubicin-treated mice or vehicle-treated mice at early metastatic stage. $n = 3$ mice per group; Scale bars, 100 μm. Top right rectangle shows 4-fold magnification. Cell nuclei-DAPI; C3-Rhodamine Red. **b** Quantification of MFI of C3 in staining shown in (**a**), a minimum of 5 fields of view per lung were assessed in $n = 3$ mice per group. *P*-values were calculated using Welch's t-test. **c** ELISA of C3a levels in lung. C3a ELISA at early metastatic stage $n = 17$ and 18 for PBS and DOX respectively (**d**), or late metastatic stage $n = 4$ and 5 for PBS and DOX respectively (**e**). **d**, **e** Bars represent range of data points, line represents mean. *P*-values were calculated using Welch's t-test. **f** Lung tissue sections from patients with breast cancer lung metastases were analyzed by IHC. Representative images of H&E staining (**g**) and C3 staining (**h**). S = stroma rich area, T = tumor cell-rich area; $n = 5$ breast cancer patients. Scale bars, 20 μm. **i** Complement score of total-lung mRNA. $n = 7$ in both groups; Bars represent range of data points, line represents mean. *P*-values were calculated using two-tailed Mann–Whitney test. **j** Correlation between complement score and micro-metastasis, quantified by mCherry mRNA expression. $r = 0.635$; $p < 0.0026$. **k** Correlation between complement score and immunosuppression score. $r = 0.6573$; $p < 0.03$. For **j**, **k** two-tailed Pearson correlation was calculated. Graphical summary was designed using BioRender. Source data are provided as a Source Data file.

antagonists and their pulmonary metastatic relapse was assessed by intravital CT imaging (Figs. 7e and 8a). Of note, complement receptor antagonists alone did not have cytotoxic effects on tumor cells (Supplementary Fig. 9h, i). Quantification of lung metastases revealed that the overall metastatic burden was not significantly different between doxorubicin-treated mice and controls (Fig. 8b). However, there were significant changes in metastatic progression and burden when doxorubicin was combined with complement receptor antagonists: treatment with C3aR or C5aR1 antagonists decreased the percentage of mice with multifocal metastases (52% vs 14% vs 6% in doxorubicin alone, C3aR and C5aR1 antagonists, respectively), while increasing the percentage of mice with unifocal metastases (4% vs 27% vs 28% in doxorubicin only, C3aR and C5aR1 antagonists, respectively). Notably, treatment with either complement receptor antagonist alone did not significantly alter the overall lung metastatic incidence (Fig. 8b). Thus, combining doxorubicin with complement receptor antagonists delayed metastatic progression and decreased the number of mice with advanced metastatic disease.

Overall, both the number of metastatic foci per mouse and the sum of metastatic area per mouse were significantly reduced in mice treated with a combination of doxorubicin and the C3aR and C5aR1 antagonists (Supplementary Fig. 9k, l). Considering the observed differences in metastatic burden among the various treatment groups, we further analyzed the differences in response to therapy among metastases-bearing mice. Analysis of metastatic load in these mice showed a marked decrease in the number and area of metastatic foci formed in lungs of mice treated with chemotherapy in combination with either C3aR or C5aR1 antagonists, compared with mice treated with doxorubicin only (Fig. 8c, d). Of note, complement antagonists treatment alone did not significantly alter the number of metastatic foci per mouse or the sum of metastatic area in lungs of treated mice, compared with controls (Supplementary Fig. 9j–l). Moreover, monitoring metastatic progression of individual mice revealed that C3aR antagonism combined with chemotherapy attenuated the progression rate of lung metastatic growth compared with controls or doxorubicin alone: Quantification of metastatic progression rate by longitudinal follow up of lesions by CT imaging, indicated that the combined treatment of doxorubicin with C3aR antagonist decelerated disease progression (Fig. 8e, f). Thus, while doxorubicin alone had adverse effects on the lung metastatic niche, combining chemotherapy with pharmacological targeting of complement signaling alleviated these immune dysregulating effects and mitigated lung metastasis. These data suggest that combining complement signaling inhibition with chemotherapy may be clinically beneficial in attenuating metastatic relapse in human breast cancer patients following resection of their primary tumor.

## Discussion

In this study, we elucidated the paradoxical effects of adjuvant chemotherapy in the microenvironment of breast cancer pulmonary metastases. We showed that systemic treatment with chemotherapy induced rapid dysregulation of the immune milieu in lungs, resulting in the formation of an immunosuppressive metastatic niche.

Following cytotoxic treatment, lung fibroblasts upregulated the expression of inflammatory mediators including complement factors, thus facilitating the recruitment of myeloid cells that modulate immunosuppression in the metastatic microenvironment. Our findings reveal that complement signaling is functionally important in mediating the adverse effects of chemotherapy, as pharmacological blockade of complement signaling in combination with chemotherapy attenuated lung metastasis (Fig. 9).

Studies in recent years provided evidence regarding the intricate host responses to conventional cytotoxic treatment and demonstrated the key role of the host immune and stromal microenvironments in mediating these responses[51]. Chemotherapeutic agents have been shown to augment tumor-promoting inflammation, restrain antitumor immunity[35,52–54], and foster a network of tumor-stromal cell interactions resulting in enhanced resistance and metastatic capability of cancer cells[55]. In this context, CAFs were suggested to promote chemoresistance of cancer cells, in addition to their tumor promoting and immune-modulating functions[56]. Indeed, a stromal gene signature in primary tumors was shown to be prognostic of chemoresistance and poor survival[10,57–59]. However, most of these findings were obtained in the primary tumor microenvironment, and the role of metastasis-associated fibroblasts in modulating the response to chemotherapy was largely unknown.

We have previously demonstrated a role for fibroblasts as sentinels of tissue damage throughout breast cancer progression, consequently prompting tumor-promoting immune responses[21,60]. In this study, we showed that systemic chemotherapy in a tumor-free setting instigated an inflammatory switch in lung fibroblasts, concurrent with a disrupted immune milieu. Importantly, these changes generated a hospitable niche in lungs that supported the colonization of breast cancer cells. Thus, inflammation induced by cytotoxic therapeutics may inadvertently potentiate metastatic progression, as suggested also by clinical findings[61].

We utilized two different mouse models of spontaneous lung metastasis following resection of a primary TNBC tumors and adjuvant chemotherapy treatment, which recapitulates the standard of care of breast cancer patients. Triple-negative breast cancer patients are generally not eligible for targeted therapy, and their treatment often depends heavily on chemotherapy. Surprisingly, using these models we found that adjuvant treatment with doxorubicin was unable to curb lung metastasis while cisplatin significantly reduced metastatic load, despite similar effectiveness of both drugs in killing cancer cells in vitro. Moreover, inspection of lungs shortly after adjuvant treatment revealed that doxorubicin accelerated the formation of early macro-metastases, while the metastatic burden at endpoint was similar to the control group. These findings highlighted the double-edged sword effect of chemotherapy[62].

We, therefore, characterized the early effects of doxorubicin therapy on the host microenvironment that enhance chemoresistance and metastatic progression. We show in a spontaneous metastasis setting that doxorubicin therapy induced rapid marked changes in the lung immune landscape manifested by a switch towards an immunosuppressive microenvironment, while cisplatin treatment enhanced T

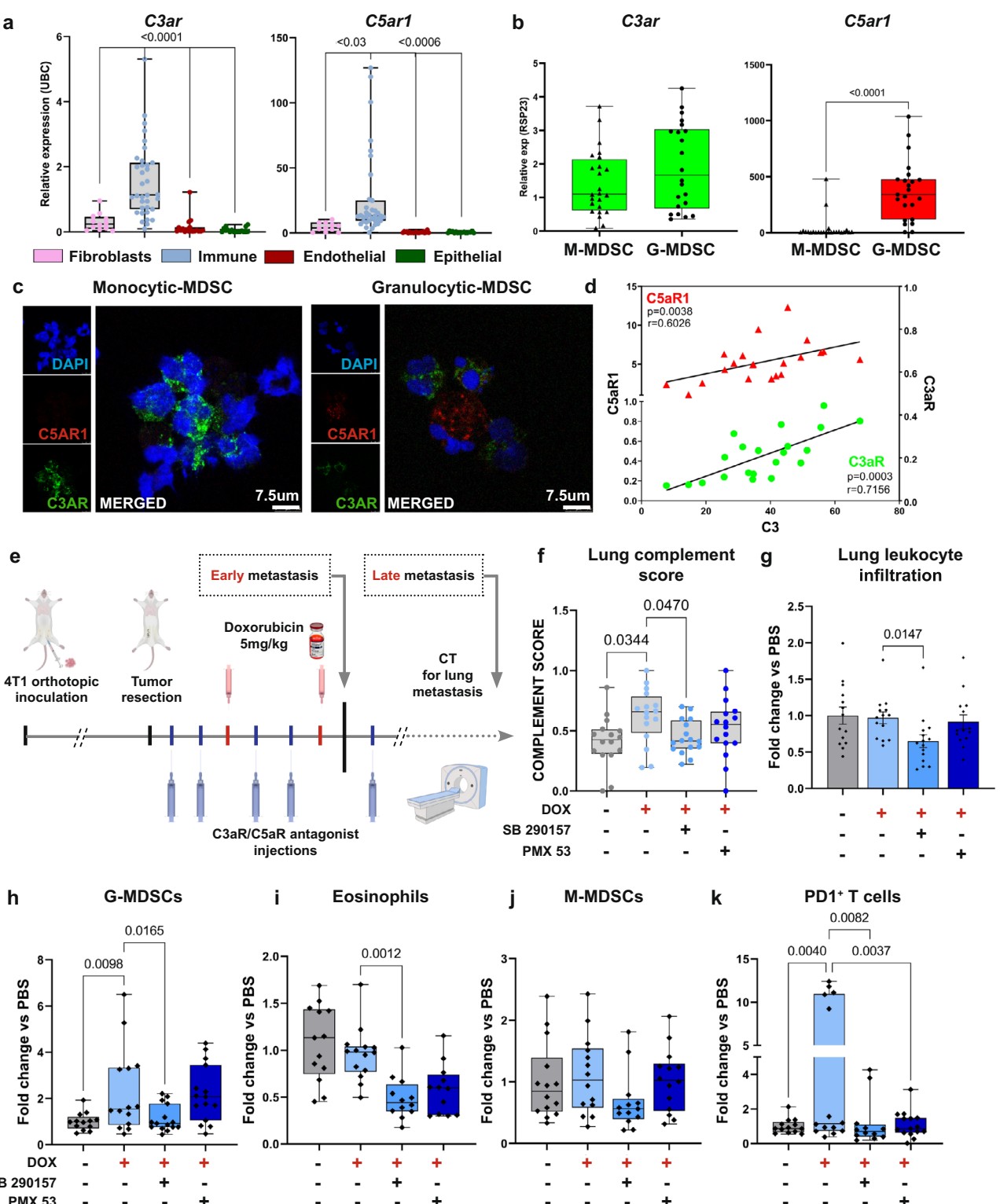

cell infiltration. These findings are in agreement with pre-clinical and clinical data of cisplatin-induced anti-tumor immune-modulation[63,64]. Indeed, a recent clinical trial showed an increase in disease-free survival and lower incidence of distant metastases in TNBC patients following platinum-based chemotherapy, compared to anthracycline-based chemotherapy[65]. Our findings suggest that the underlying mechanism for these discrepancies in therapy efficiency may be that doxorubicin, but not cisplatin, upregulates complement signaling and enhances MDSCs recruitment to the metastatic niche which facilitates

metastatic growth. Conversely, cisplatin increases T cell infiltration, supporting anti-tumor immunity and curbing the formation of an immunosuppressive metastatic niche.

Adjuvant doxorubicin enhanced the recruitment of MDSCs to the lung metastatic niche, and led to boosting of the immunosuppressive phenotype of these cells, thus limiting anti-tumor immunity by enhancing T cell dysfunction. Chemotherapy-induced infiltration of MDSCs was previously observed in mouse models of primary tumors[66,67]. Our findings in a spontaneous metastatic setting suggest

**Fig. 7 | Combining chemotherapy with blocking of complement signaling attenuates dox-induced immune modulation. a** qRT-PCR analysis of the anaphylatoxin receptors *c3ar* and *c5ar1* expression in sorted lung cell populations (endothelial cells, epithelial cells, immune cells and fibroblasts). *n* = 35; 11; 20 and 17 for CD45⁺, PDGFRα⁺, CD31⁺ and EpCAM⁺ respectively. Bars represent range of data points, line represents mean in relative expression to housekeeping gene (*ubc*); *P*-values were calculated using one-way ANOVA followed by Tukey multiple comparisons test were performed. **b** *c3ar* and *c5ar1* expression in granulocytic-MDSCs and mono-MDSCs isolated from lungs. *n* = 25 and 23 for mMDSCs and gMDSCs. Bars represent range of data points, line represents mean in relative expression to housekeeping gene (*rsp23*); *P*-values were calculated using two-tailed Mann–Whitney test. **c** Immunofluorescence of C3aR-FITC and C5aR1-PE in M-MDSCs (left) and G-MDSCs (right) sorted from lungs, and placed on microscope slides by cytospin. DAPI-blue. Scale bar, 7.5 μm. Data is representative of one independent experiment with *n* = 5. **d** Two tailed Pearson correlation between lung mRNA levels of *c3* with *c3ar* (green, *r* = 0,7156, *p* = 0,0003) or *c5ar1* (red, *r* = 0,6026, *p* = 0.0038). **e** Experimental scheme: following orthotopic injection of 4T1 cells and

primary tumor resection, mice were treated with doxorubicin (5 mg/kg) in combination with C3aR antagonist (SB290157- 10 mg/kg) or with C5aR1 antagonist (PMX53- 1 mg/kg), or vehicle (administered i.p). Lung metastasis was monitored by CT imaging. **f** Complement score calculated from total lung mRNA at early metastatic stage, 24 h following treatment with last dose. *n* = 16 in each group. Bars represent range of data points, line represents mean. *P*-values were calculated using Brown–Forsythe ANOVA multiple comparison test. **g–k** Flow cytometry analysis of lung infiltrating immune cells at early-metastatic stage. **g** Total immune cells (CD45⁺ cells), data presented as mean ± s.e.m (**h**) G-MDSCs (CD45⁺CD11b⁺SiglecF⁻Ly6G⁺Ly6Cⁱⁿᵗ), **i** Eosinophils (CD45⁺CD11b⁺Ly6G⁻CD11c⁻SiglecF⁺), **j** Mono-MDSCs (CD45⁺CD11b⁺SiglecF⁻Ly6G⁻Ly6C⁺), **k** PD1⁺ T cells. *n* = 14, 14, 15 and 15 mice in PBS, DOX, DOX + SB290157, and DOX + PMX53, respectively. **h–k** Bars represent range of data points, line represents mean. **g–k** *P*-values were calculated using Kruskal–Wallis test for multiple comparisons. Data are representative of three independent experiments and are presented as mean normalized to control. Graphical summary was designed using BioRender. Source data are provided as a Source Data file.

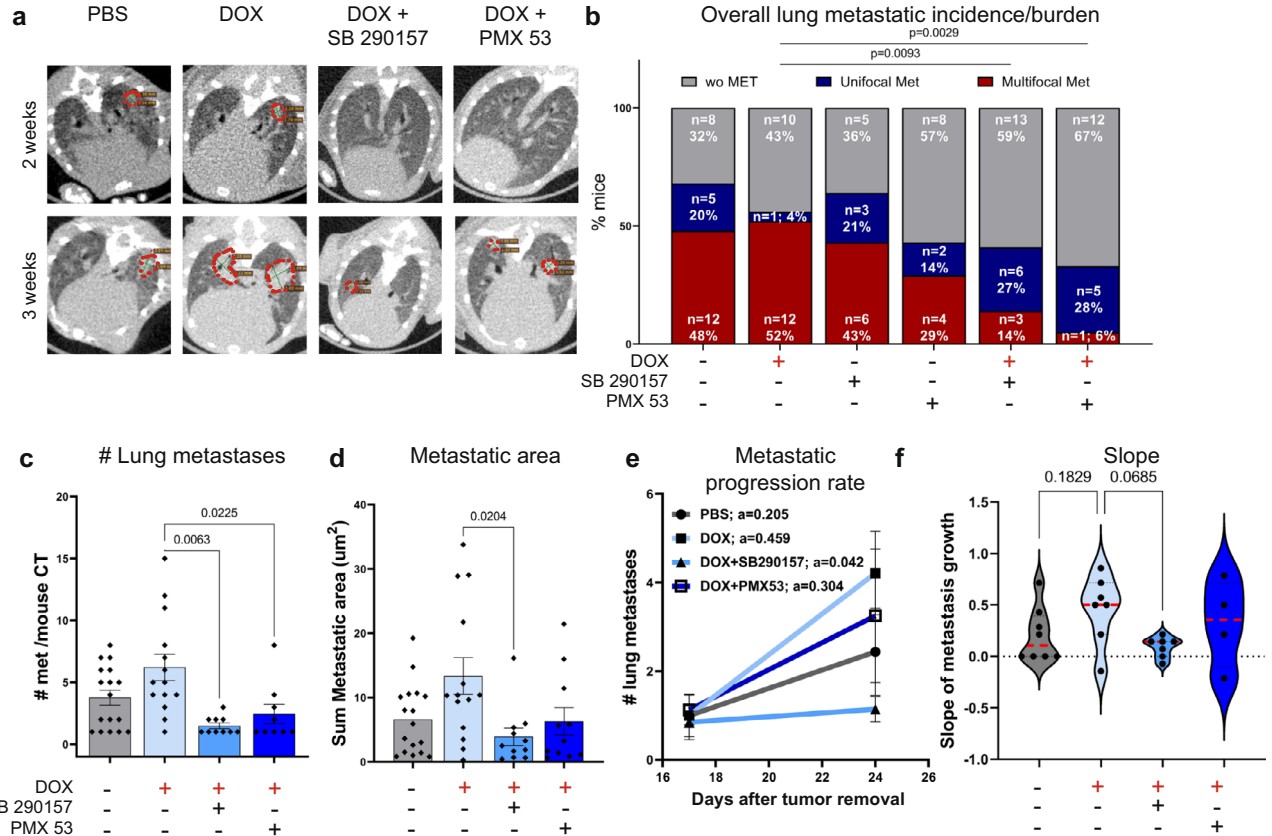

**Fig. 8 | Combining chemotherapy with blocking of complement signaling inhibits lung metastatic relapse. a** Representative CT images of lungs, 2 and 3 weeks after resection of 4T1 primary breast tumor in control group (PBS), doxorubicin-treated mice (DOX), combination of doxorubicin and C3aR antagonist (DOX + SB290157) or combination of doxorubicin and C5aR1 antagonist (DOX + PMX53); administered i.p. Metastatic lesions are circled in red. **b** Quantification of overall incidence of lung metastasis as analyzed by CT imaging in all groups. *n* = 25, 23, 14, 14, 22 and 18 mice in PBS, DOX, SB290157, PMX53, DOX + SB290157, and DOX + PMX53, respectively. Data are presentative of 2 and 3 independent experiments, for antagonist-only groups and for combination therapy groups, respectively. *P*-values were calculated using two-tailed Chi-square test. **c, d** Assessment of

metastasis burden determined by: number of lung metastatic foci *n* = 17, 14, 10, and 9 mice in PBS, DOX, DOX + SB290157, and DOX + PMX53, respectively. **c** Metastatic area *n* = 17, 14, 11, and 11 mice in PBS, DOX, DOX + SB290157, and DOX + PMX53, respectively **d** in metastases-bearing mice treated as indicated. Data are presented as mean ± s.e.m. *P*-values were calculated using Kruskal–Wallis test for multiple comparisons. **e** Metastatic progression rate. "a" represents line slope. **f** Slope of (**e**), each point represents a mouse. Red dashed lines represent the median slope value. **e, f** *n* = 8, 7, 7 and 4 mice in PBS, DOX, DOX + SB290157, and DOX + PMX53, respectively. Error bars represent s.e.m. *P*-values were calculated using Kruskal–Wallis test for multiple comparisons. Source data are provided as a Source Data file.

that systemic effects of adjuvant therapy may nurture an immunosuppressive metastatic niche. The shift towards an immunosuppressive microenvironment following doxorubicin treatment may have important clinical implications regarding the potential benefit of

combining immunotherapy with chemotherapy in regular or metronomic regimens, as recently reported[68,69].

We show that lung fibroblasts responded to chemotherapy by upregulating the expression of complement factors, known to play

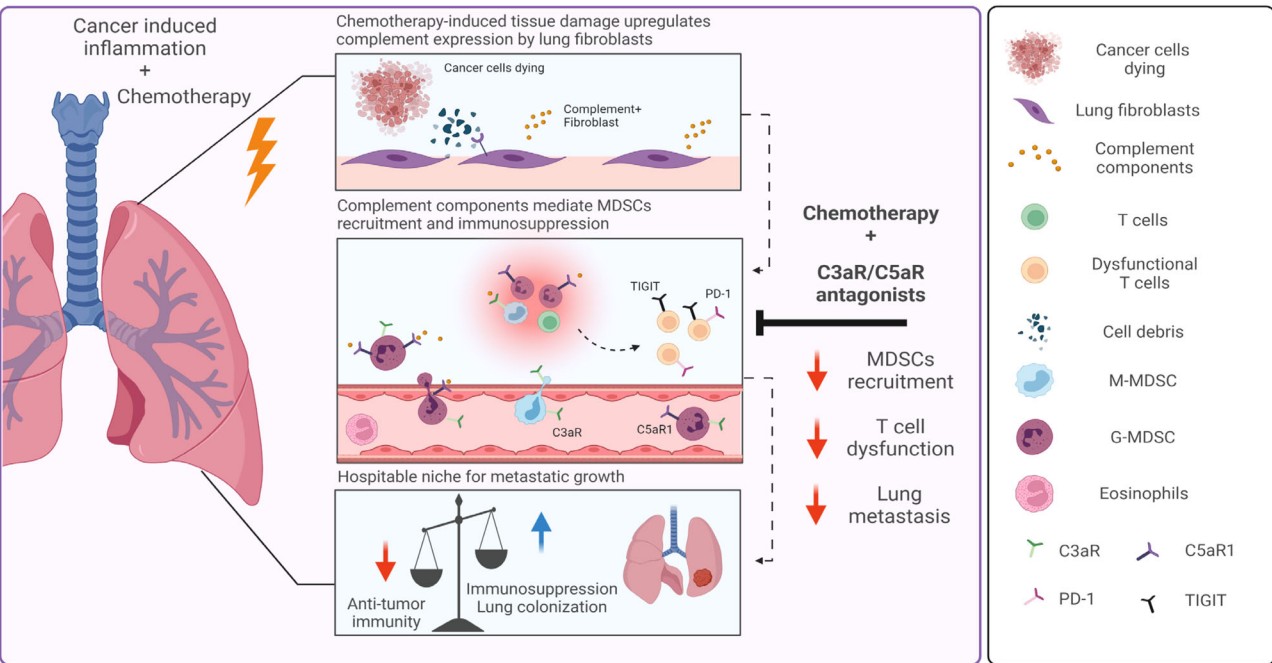

**Fig. 9 | Chemotherapy-induced complement signaling in fibroblasts modulates immunosuppression and metastatic relapse in breast cancer.** Adjuvant doxorubicin treatment following primary tumor removal instigated complement signaling in lung fibroblasts, leading to recruitment of MDSCs and the formation of an immunosuppressive niche, favorable to metastatic relapse. pharmacological targeting of complement signaling in combination with chemotherapy alleviated immune dysregulation and attenuated lung metastasis. Graphical summary was designed using BioRender.

a role in tumor-promoting inflammation[30,70]. CAFs were recently implicated as mediators of complement signaling in cancer[57,71]. In particular, fibroblast-derived C3a was shown to be instrumental in mediating infiltration of tumor-promoting macrophages to melanoma tumors[33]. In this study, we show that fibroblasts in the lung metastatic niche were the main source of C3a and other complement factors following systemic chemotherapy. Moreover, we demonstrate that pharmacological targeting of C3a-C3aR signaling in combination with adjuvant doxorubicin treatment reversed chemotherapy-induced immunosuppressive modulation of the immune microenvironment and attenuated lung metastasis. Pharmacological blockade of complement pathway was previously shown to inhibit MDSCs recruitment to tumors and to reduce metastasis by altering T cell responses in a mouse model of breast cancer[48,72]. Moreover, recent studies suggested that C3aR and C5aR1 function as immune-checkpoint receptors or immune-regulators[73,74]. Our findings functionally link chemotherapy-induced upregulation of complement signaling with the formation of a hospitable metastatic niche in lungs, and support the potential of targeting complement signaling as a beneficial combination treatment strategy to combat metastatic relapse.

Importantly, these findings need to be further investigated in human breast cancer patients. However, obtaining human samples of lung tissues in temporal proximity to chemotherapeutic administration is logistically unfeasible. Unfortunately, blood samples as surrogate for assessing the role of complement signaling in lungs following chemotherapy may not be sufficiently informative: conflicting reports regarding plasma levels of complement components in cancer patients receiving chemotherapy complicates the interpretation of circulation levels of complement components in the context of prognostic assessment[75,76].

In summary, our study highlights the complexity in the balance between the beneficial and adverse effects of chemotherapy, caused by tissue damage and inflammation. Our findings suggest that blockade of complement signaling in combination with adjuvant chemotherapy in breast cancer patients may attenuate the adverse effects of doxorubicin treatment, thus offering a promising approach for clinical use.

## Methods

### Mice
All experiments involving animals were approved by the Tel Aviv University Institutional Animal Care and Use Committee (IACUC). Animals were maintained in specific pathogen-free conditions with controlled temperature/humidity (22 °C/55%) environment on a 12 h light-dark cycle and with food and water ad libitum. All animals were maintained within the Tel Aviv University Specific Pathogen Free (SPF) facility. All in vivo experiments were performed using 6–8 weeks old female BALB/c or C57BL/6 mice (Harlan, Israel), unless otherwise stated. No statistical method was used to pre-determine mouse sample size, but the sample size was chosen to be adequate to receive significant results as determined by preliminary experiments. Mice used in experiments were not randomized, and the investigators were not blinded to allocation during experiments and outcome analysis. Mice that died for unknown reasons were excluded from analysis. Mice in early-stage metastasis experiments were euthanized 24 h after the second dose of chemotherapy or vehicle. Mice in late-stage metastasis experiments were euthanized when more than 80% of control mice developed macro-metastases detected by CT imaging. Tumors were removed at an early stage, and therefore did not reach the 15 mm maximal permitted tumor burden. All experiments were repeated independently at least twice.

### Cancer cells
4T1 cells were received from Dr. Zvi Granot. EO771 cells were received from Dr. Ruth Scherz-Shouval. Cells were cultured with RPMI 1640 medium supplemented with 10% FCS, 1% penicillin-streptomycin, 1% sodium-pyruvate, 1% HEPES 1 M and 0.5% glucose and maintained at 37 °C with 5% $CO_2$. Cells were routinely tested for mycoplasma using the EZ-PCR-Mycoplasma test kit (Biological Industries; 20-700-20) and confirmed negative.

## Primary lung fibroblasts isolation and culture

Lungs were isolated from 6 to 8 weeks old BALB/c female mice and dissociated. Briefly, lung tissues were harvested, washed in PBS, minced thoroughly with scissors and incubated for 40 min with RPMI supplemented with 0.1% collagenase IV (Worthington, LS004177) and 0.1% diapase II (Roche, 04942078001) on stir plate in 37 °C water bath. Red blood cells were lysed and remaining single cell suspensions were seeded on 6-well plates pre-coated with Rat tail collagen (Corning; 354236). Primary fibroblasts were grown in RPMI1640 supplemented with 10% FCS, 1% penicillin-streptomycin and 1% Sodium-pyruvate.

## Cell conditioned media preparation

For 4T1-CM: when cells reached 80% confluency, plates were washed twice with PBS, and fresh serum-free medium (SFM) was applied. After 48 h, medium was collected, filtered through 0.2 µm filters under aseptic conditions, flash-frozen in liquid nitrogen, and stored at −80 °C. SFM was used as control.

For DAMP-rich CM: 1 µM Doxorubicin/Cisplatin was added to 80% confluent cultures. 24 h later, cells were washed, and fresh SFM was applied. After an additional 24 h, medium was collected and filtered as described above.

## Lung homogenate supernatant

Lungs were perfused with 10 ml PBS before harvesting. Perfused lungs were placed in 70 µm cell strainers in 50 mL tube containing RPMI1640 media. Lungs were homogenized using a syringe plunger, and centrifuged 7 min at $500 \times g$. Supernatants were collected and filtered through 0.45 µm filters.

## Enzyme-linked immunosorbent assay (ELISA)

C3a ELISA was performed for lung homogenate supernatant using Mouse Complement Component 3a ELISA Kit, according to manufacturer's instructions (FineTest. EM0882).

## Immunostaining

**Tissue sections.** Lungs were injected intra-tracheal with 600 µL Optimal Cutting Temperature compound (O.C.T, BN62550, Tissue-Tek), harvested, shortly washed in PBS, and embedded in O.C.T on dry ice. Serial sections were obtained to ensure equal sampling of the examined specimens (10 µm trimming).

**Immunofluorescence.** Tissue sections were incubated overnight at 4 °C with Rat anti-mouse C3 antibody (11H9-ab11862). Sections were washed with PBS and incubated with a RedX−conjugated donkey anti-Rat secondary antibody (1:200, 712-035-153, Jackson ImmunoResearch Laboratories) for 2 h at room temperature, followed by DAPI (1:2000, Molecular Probes; D3571). Slides were mounted with VECTASHIELD® HardSet antifade (VE-H-1400, Vector Laboratories), visualized using the Leica Aperio VERSA slide scanner and analyzed with the ImageScope software. Quantitative analyses were performed using ImageJ Software.

**Cytospin staining.** MDSCs were seeded on coated slides for Cytospin. Cells were then fixated for 10 min with PFA 4%, washed, and incubated for 2 h at room temperature with C3aR-FITC antibody (C3aR (74): sc-53785) and C5aR1-PE antibody (Biolegend − 135806), washed and incubated with DAPI (1:2000, Molecular Probes; D3571). Slides were and mounted as above.

## Human C3 staining

Human patient samples ($n = 5$) were collected with written informed consent and processed at the Sheba Medical Center, Israel, in accordance with recognized ethical guidelines, under an approved Institutional Review Board (3112-16). Tissue sections stained for C3 (LS-B4290, LSBio) were analyzed by an expert pathologist. Images

were scanned at ×20 magnification using the Leica Aperio VERSA slide scanner.

## T cell suppression assay

Splenocytes were isolated BALB/c mice, and labeled with CFSE (5 M, Biolegend, 423801). $1 \times 10^5$ cells in 100 µL per well were plated in a 96-well plate pre-coated with anti-mouse CD3ε (1 µg/mL, SouthernBiotech, 1530-01). CD11b⁺Ly6C$^{int}$Ly6G⁺ cells (G-MDSCs) or CD11b⁺Ly6G⁻Ly6C⁺ cells (M-MDSCs) were isolated from spleens of doxorubicin-treated mice or controls using FACS-sorting, and incubated with stimulated T cells ($5 \times 10^4$ MDSC/well). Dilution of CFSE was evaluated 3 days later by flow cytometry.

## Flow cytometry analysis

Single cell suspensions were prepared according to organ (circulating cells were isolated, lungs were enzymatically digested, spleens were mechanically suspended). Cells were filtered by 70 µM cell strainers (Corning); red blood cells were lysed. Cells were counted and resuspended in FACS buffer (PBS with 2% FCS and 2 mM EDTA). Cells were incubated with anti-mouse CD16/CD32 (eBioscience, 16-0161-82, dilution 1:100), followed by staining with the following anti-mouse antibodies: anti-CD45-BV650 (BioLegend, BLG-103151, dilution 1:100); anti-CD11b-PeCy7 (BioLegend, BLG-101215, dilution 1:100); anti-CD11c-PerCP-Cy5.5 (eBioscience, 45-0114, dilution 1:100); anti-SiglecF-APC-R700 (BD Biosciences, BD565183, dilution 1:100); anti-Ly6G-APC (BioLegend, 127614, dilution 1:200); anti-Ly6C-FITC (BioLegend, 128006, dilution 1:200); anti-NKp46-PeCy7 (BioLegend, BLG-137617, dilution 1:100); anti-B220-PerCP-Cy5.5 (BioLegend, BLG-103235, dilution 1:100); anti-CD4-APC-Cy7 (BioLegend, BLG-100413, dilution 1:100); anti-CD8a-APC (BioLegend, BLG-100712, dilution 1:100); anti-CD8a-PE (eBioscience, 12-0083, dilution 1:100); anti-CD3-FITC (BioLegend, BLG-100306, dilution 1.5:100); anti-PD-1-BV785 (Biolegend, BLG-135225, dilution 1:100); anti-TIM3-BV605 (Biolegend, BLG-119721, dilution 2:100); anti-LAG3-PerCP-Cy5.5 (Biolegend, BLG-125212, dilution 1:100); anti-TIGIT-APC (Biolegend, BLG-142106, dilution 4:100) and DAPI (Molecular Probes; D3571). The specificity of staining was validated by appropriate fluorescence minus one (FMO) method. Immune populations were defined as described in Supplementary Fig. 2a, b and Supplementary Fig. 6d. Analysis was performed with CytoFLEX Flow Cytometer (Beckman Coulter, Inc.) and data analysis was done with FlowJo Software (version X.0.7).

## FACS sorting

Single cell suspensions were stained with cell type-specific antibodies indicated above. DAPI was used to exclude dead cells (1:2000 Molecular Probes; D3571). Sorting was performed using BD FACSAria III to isolate lung fibroblasts (CD45⁻CD31⁻EpCAM⁻PDGFRα⁺); endothelial cells (CD45⁻EpCAM⁻PDGFRα⁻CD31⁺); epithelial cells (CD45⁻CD31⁻PDGFRα⁻EpCAM⁺) and immune cells (EpCAM⁻CD31⁻PDGFRα⁻CD45⁺). Gating strategy is described in Supplementary Fig. 6h. Cells were incubated with anti-mouse CD16/CD32 (eBioscience, 16-0161-82), followed by staining with the following anti-mouse antibodies: anti-CD45-PerCP-Cy5.5 (eBioscience, 45-0451, dilution 1:100); anti-CD31-PeCy7 (eBioscience, 25-0311, dilution 1:100); anti-EpCAM-APC (eBioscience, 17-5791, dilution 1:100) and anti-PDGFRa-PE (eBioscience, 12-1401-81, dilution 1:100). Granulocytic and Monocytic MDSCs were isolated as CD11b⁺Ly6C$^{int}$Ly6G⁺ and CD11b⁺Ly6G⁻Ly6C⁺, respectively. Gating strategy is described in Supplementary Fig. 2a. Cells were incubated with anti-mouse CD16/CD32 (eBioscience, 16-0161-82), followed by staining with the following anti-mouse antibodies: anti-CD45-BV650 (BioLegend, BLG-103151, dilution 1:100); anti-CD11b-PeCy7 (BioLegend, BLG-101215, dilution 1:100); anti-CD11c-PerCP-Cy5.5 (eBioscience, 45-0114, dilution 1:100); anti-SiglecF-APC-R700 (BD Biosciences, BD565183, dilution 1:100); anti-Ly6G-APC (BioLegend, 127614, dilution 1:200); anti-

Ly6C-FITC (BioLegend, 128006, dilution 2:100). Sorts were performed using FACSDiva software v8.

## Lung colonization assay

8 weeks old naïve BALB/c mice were injected i.p with Doxorubicin (5 mg/kg), Cisplatin (5 mg/kg) or PBS, 5 days and 2 days prior to intravenous (i.v) injection of $1 \times 10^5$ cells 4T1-luc cells. Lung colonization was analyzed by IVIS imaging for luciferase activity.

## RNA isolation and qRT-PCR

RNA from sorted cells was isolated using the EZ-RNAII Kit (20-410-100, Biological Industries). RNA from in vitro experiments and from total lungs was isolated using the PureLink RNA Mini Kit (Invitrogen; 12183018 A). RNA samples were analyzed with NanoDrop 2000c Spectrophotometer. cDNA synthesis was conducted using qScript cDNA Synthesis Kit (Quanta, 95047-100). qRT-PCR were performed with PerfeCTa SYBR Green Fastmix ROX (Quanta, 95073-012). Expression results were normalized to Gusb, Gapdh, rsp23 flat2 or Ubc and to controls. RQ (2 − ΔΔCt) was calculated. Primers sequences used are detailed in Supplementary Table 1.

## Orthotopic tumors transplantations and adjuvant chemotherapy treatment

**Early metastatic stage experiment.** $5 \times 10^5$ 4T1 or EO771 cells were suspended in PBS and mixed 1:1 with Matrigel (BD Biosciences, 354230). 100 μl of cell mixture was injected into the fat-pad of the right inguinal mammary glands of 8 weeks old female BALB/c or C57BL/6 mice depended on the cell type. Tumors were resected 2 weeks following injection. 4 days after tumor removal, first dose of chemotherapy was injected; followed by a second dose 5 days later. Mice were euthanized 24 h after the 2nd dose.

**Late metastatic stage experiment.** $2 \times 10^5$ 4T1 cells, or $5 \times 10^5$ EO771 cells were injected as above. Primary tumors were resected 3 weeks following injection. Chemotherapeutic treatment was administered as above. Lung metastases were monitored by CT imaging.

## Metastasis quantification

**H&E quantification.** Lung tissue sections were stained with H&E using Sakura Tissue-Tek Prisma (Department of Pathology, Tel Aviv Sourasky Medical Center). Quantification of lung metastatic load was performed by analyzing the number of metastatic lesions per section or by evaluating the metastatic area per section. At least three regions of each mouse were analyzed. Images were obtained at ×20 magnification using the Leica Aperio VERSA slide scanner and analyzed with the ImageScope software.

**CT imaging quantification.** Lung metastases were monitored by CT imaging starting 10 days after primary tumor removal. Metastasis quantification included counting the total number of metastases and measurement of metastatic area.

## Scratch assay

Normal lung fibroblasts (NLFs) were pre-conditioned with doxorubicin (1uM), cisplatin (5uM), or serum-free media (SFM) for 24 h. Treated NLFs were plated in a 96-well IncuCyte imageLock plate (Essen BioScience). A scratch was introduced using the IncuCyte Wound-Maker (Essen Bioscience). Wells were washed with PBS and cytotoxic treatment or SFM were applied for 48 h in the IncuCyte system (Essen BioScience). Images were analyzed using the IncuCyte software.

## Collagen contraction assay

NLFs were pre-conditioned as above. $1.5 \times 10^5$ fibroblasts were suspended in a mixture of SFM mixed with High Concentration Rat Tail Collagen, type 1 (BD Biosciences) and allowed to set at 37 °C for 45 min,

followed by incubation for 24 h. Gels were imaged at endpoint and analyzed with ImageJ software to assess gel area.

## 4T1 killing assay

**PI incorporation of dead cells.** 4T1 were plated in in a 96-well Incu-Cyte imageLock plate (Essen BioScience). Cells were starved overnight with SFM followed by incubation with Doxorubicin (5 μM), Cisplatin (5 μM) or SFM for 48 h. PI incorporation was analyzed with the Incu-Cyte system (Essen BioScience).

**XTT assay.** XTT assay (Biological Industries, 20-300-1000) was performed 24 and 48 h following incubation with cytotoxic drugs as above according to the manufacturer's instructions.

## Lung fibroblast transcriptome analysis

**RNA-seq.** CD45⁻EpCAM⁻CD31⁻PDGFRα⁺ fibroblasts were isolated by cell sorting from early stage metastatic lungs of mice treated as described above. MARS-seq protocol was used to generate libraries[77]. MARS-seq libraries were sequenced using Illumina NovaSeq 6000. Sample barcodes were extracted from read 2 and concatenated to the fastq header of read 1. Reads were mapped to the Mus musculus genome (mm10) using STAR. Read counts per gene were calculated using HTSeq-count and a Refseq gtf file. Gene set enrichment analysis was performed using the GSEA Java Desktop tool (v4.1.0). Normalized gene expression levels were enrolled into a pre-defined dataset of the complement pathway (REACTOME_COMPLEMENT_CASCADE, source R-HSA-166658) using the following parameters: 100 permutations, gene-set permutation, tTest statistical test was used.

**nCounter gene expression profiling.** Lung fibroblasts from early-stage metastatic lungs were sorted and RNA was isolated as described above and analyzed with nCounter Mouse Inflammation v2 Panel, providing gene expression analysis for 248 inflammation-related mouse genes (and 6 internal reference controls), (NanoString Technologies). Raw data were normalized to internal reference genes using the nSolver software, following the manufacturer's instructions.

## Single cell RNA-seq analysis of public databases

ScRNA-seq data of human lung adenocarcinoma and lung stroma of mice with lung metastases were retrieved from Gene Expression Omnibus (GSE123902 and GSE149636, respectively). Seurat v2.3.4 was used under R v3.6.0 to filter cells by the number of genes expressed, the number of unique molecular identifiers and the percentage of mitochondrial genes expressed, cluster and visualize cells by similarity in gene expression and identify cluster markers. Briefly, Seurat objects were created and cells were removed from analysis if they had unique feature counts >4000 or <200, or if mitochondrial counts were >5%. Counts were normalized and scaled. Clustering and 2D projection by t-distributed Stochastic Neighbor Embedding (t-SNE) and Uniform Manifold Approximation and Projection (UMAP) were performed after dimensional reduction using the first 15–20 Principal Components (PCs). Plots were generated using the ggplot2 function.

Processed single-cell RNAseq data of lung adenocarcinoma and lung squamous cell carcinoma tumors were retrieved from ref. 78. GraphPad Prism software was used to generate heatmaps depicting expression of cluster-specific genes and complement-related genes.

## In vivo blockade of complement signaling

Mice were treated with doxorubicin alone as above, or with doxorubicin (5 mg/kg) supplemented with either C3aR (10 mg/kg per injection, SB 290157 (trifluoroacetate salt) - Cayman chemicals) or C5aR1 (1 mg/kg per injection, PMX53 - Calbiochem) antagonists. Two days following tumor resection, mice were injected intraperitoneally with either of the complement receptor antagonists, or with vehicle. Injections were performed every other day during the time-period

between the first and second chemotherapy doses, for a total of four (for early metastatic stage) or five (for late metastatic stage) injections.

### Graphical illustrations
Graphical elements used to create experimental design schemes were created using BioRender.

### Statistical analysis
Statistical analyses were performed using GraphPad Prism software. For two groups, statistical significance was calculated using t-test with Welch correction unless otherwise stated. For more than two comparisons, One-Way ANOVA with Tukey correction for multiple comparisons was applied unless otherwise stated. $P$-value of ≤0.05 was considered statistically significant unless otherwise stated. All experiments represent at least three biological repeats. Correlation analysis was performed using Pearson correlation, $p$-value of ≤0.05 was considered statistically significant. Outliers were identified and removed using Grubbs' or ROUT method.

### Reporting summary
Further information on research design is available in the Nature Research Reporting Summary linked to this article.

## Data availability
Publicly available scRNA-seq data of human lung adenocarcinoma and lung stroma of mice with lung metastases were retrieved from Gene Expression Omnibus: GSE123902; GSE149636; Nanostring nCounter data and RNAseq data in support of this study are deposited at the Gene Expression Omnibus (GEO) with accession numbers GSE207990 and GSE208289, respectively:

The remaining data are available within the Article, Supplementary Information or Source Data file. Source data are provided with this paper.

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

## Acknowledgements

The authors thank Dr. Irena Shur and Dr. Daria Makarovsky at the Faculty of Medicine Interdepartmental Core Facility (SICF) for their help with imaging and FACS analyses and DAntes Design for some of the graphical illustrations used. This study was supported by grants to N. Erez from the European Research Council (ERC StG, 637069 MetCAF), the Israel Cancer Association (ICA), and The Emerson Collective Cancer Research Fund. N. Erez and AS received supported by the Richard Eimert Research Grant on Solid Tumors.

## Author contributions

L.M., N. Ershaid, and N. Erez conceptualized and designed the study. L.M., N. Ershaid. performed experiments and analyzed the data. H.D., Y.Z. participated in data analysis and interpretation. S.B.-Y. performed collagen contraction and scratch assays, overseen by N. Ershaid. C.A. and I.B. performed and analyzed human patient staining, L.M., N. Ershaid, and N. Erez wrote the manuscript. A.S., Y.S., Y.Z., and H.D. reviewed and edited the manuscript. N. Erez supervised the study.

## Competing interests

The authors declare no competing interests.
