## [Peer Review File · Nature Communications]

Chemotherapy-induced complement signaling modulates immunosuppression and metastatic relapse in breast cancerREVIEWER COMMENTS

Reviewer #1 (Remarks to the Author):

The manuscript by Monteran et al., describes work showing that doxorubicin, but not cisplatin impacts infiltrating tumor immune cells in the lung and that doxorubicin causes lung fibroblasts to expression complement factors C3aR and C5aR1 that they argue increases metastasis. The observations are interesting and experiments are well controlled. In addition, this is an important area of inquiry. An outstanding question, that isn't addressed in the work is whether expression of C3aR and C5aR1 and the associated immune changes are related and more importantly if those changes are responsible for the changes in metastases that is claimed. Of equal importance, it isn't entirely clear to this reviewer if the metastatic changes that are claimed to occur upon treatment with the C3aR or C5aR1 agonists are significant; is there an overall reduction in metastatic tumor burden. These issues should be directly addressed. Below I provide several points that should be addressed in no particular order of importance.

1. In Figure 1 there are changes reported as a percent of CD45. Do total CD45 cell numbers go up as well or is it only a change in distribution?
2. The authors suggest that colonization is higher in chemo treated mice compared to PBS treated mice but when one looks at the BLI images they honestly don't look different if one considers the large variation. That being said, there are clear differences at later timepoints. To me that suggests that colonization may not be different but rather survival of seeded cells and/or proliferation is greatly enhanced in the chemo treated mice. Did the authors do BrdU and/or caspase stains early or could they address this point in any other way?
3. In Figure 1 the authors argue that doxorubicin and cisplatin induced similar inflammatory/fibrosis signatures and in the seeding assay, both increased tumor outgrowth. In Figure 2 when examining spontaneous metastasis following resection, cisplatin reduces metastasis while doxorubicin does not. What is the disconnect in the two assays? Could this be a dosing effect, in other words was the maximal tolerated dose used in both chemotherapies (i.e., I know that 5 mg/kg is not the maximum dose for doxorubicin but I do not know if that is true for cisplatin)? This seems to be an important observation that should be discussed and a dose course might be helpful here.
4. Figure 2D shows metastatic area, is there a difference between PSB and doxorubicin? Based on the argument put forward, there should be greater metastasis in doxorubicin but no p values is provided. This needs to be explained/addressed.
5. Figure 3B shows increased metastasis 24 hours after doxorubicin compared to PBS, but that does not equate to increased metastasis (Figure 2) at what looks to be one week after the last doxorubicin (Figure 2A but the day isn't listed on the figure, it should be). What does this early increase mean if it does not equate to increased metastasis?
6. Figure 3 shows fold changes in immune populations, it would be more informative to show percent CD45 (main figure) and total cell counts (in supplementary) to provide more insight into what these changes really look like.
7. Figure 4 addresses the functional state of myeloid cells following chemotherapy and so compares PBS versus doxorubicin treated mice. The setup was that doxorubicin is different than cisplatin so it would be nice to show there are no suppressor, or reduced suppressor cells in cisplatin lungs immediately after treatment (again, this after showing both chemotherapies are at their maximal doses).
8. Figure 4F and beyond, as stated earlier, it would be more informative to shown percent of CD45 and total numbers opposed to fold changes, are significance values maintained?
9. Figure 4G, T cell suppression assays are typically presented with dilutions to truly assess the suppressive nature and a control with no "suppressor" added to assess maximal T cell proliferative capacity in the assay. These should be added to the figure.
10. Gating strategies for all flow should be provided in the supplementary data so the reviewer/reader can evaluate the FACS data. Figure 4J is looking at checkpoint expression and there isn't an obvious population, just a shift, which is not a surprise per se but evaluation of the gating would provide more confidence in the differences being reported.
11. Suppl Figure 3, the PCA plot shows that one of the doxorubicin treated mice is quite distinct from

the other two, could the authors discuss the variation they observe?

12. They state that they are comparing their single cell murine data to human metastatic single cell data (GSE14020) in Supple Figure 4. This is array data of metastasis and this should be clarified as much of the signal could (or may not) come from tumor cells.

13. They have compared and contrasted cisplatin to doxorubicin. In Figure 5, if cancer cells are treated with cisplatin and that is used on lung fibroblasts, what happens to complement expression? This seems to be an important control.

14. Only the C3aR antagonist reduced immune infiltration while both the C3aR and C5aR1 antagonist reduced metastasis in Figure 8, does this suggest it is not the immune infiltrate that is responsible for the differences in metastasis? This seems contrary to their hypothesis.

15. In Figure 8 they argue that the number of metastases are reduced when a C3aR or C5aR1 agonist is combined with doxorubicin but what happens to overall burden, is it reduced, if it is not, this is not that impressive.

16. In regard to Figure 8, it is important to show the tumor burdens when mice are treated with C3aR or C5aR1 agonist alone on the same graphs --- currently these data are in the supplemental and it looks like the C5aR1 agonist may change lung burden and C3aR agonist may impact lung number but is it impossible to tell with them on different graphs and the axes are different.

17. Also Figure 8, I know what a slope is, but it would help if the authors described how this was calculated for these data.

Reviewer #2 (Remarks to the Author):

In this manuscript Monteran and colleagues have addressed a highly clinically relevant question - namely what are the adverse effects of chemotherapy on normal tissue damage, how this impacts metastatic colonization and are there approaches that can be taken to counteract the adverse effects of chemotherapy whilst retaining its anti-tumor effects.

The authors have used the 4T1 model of breast cancer metastasis in syngeneic BALB/c mice. They clearly demonstrate that anthracycline chemotherapy (doxorubicin or cisplatin) clearly impact the normal lung tissue (changes in fibroblasts gene expression, changes in immune cell populations) - observations that in themselves are not novel but set the scene for their subsequent studies. What is interesting is that the different chemotherapies have different effects and the authors provide evidence (see comments below) for increased metastatic outgrowth in mice pretreated with chemo and then inoculated intravenously with 4T1 cells. For the majority of subsequent experiments, the authors predominantly use a model whereby 4T1 primary tumors are resected and then the mice treated with chemo - this is to be commended as it mimics the treatment scenario for the majority of breast cancer patients (although increasingly patients with TNBC are treated with neo-adjuvant chemotherapy). Again the authors show changes in the immune compartments following early and later stage metastatic colonization, and in the transcription profile of isolated lung fibroblasts, particularly of the upregulated expression of complement genes. Finally, the key experiment, the authors demonstrate that compared to mice treated with doxorubicin alone, combination treated with complement receptor antagonists reduces metastatic outgrowth in the lungs.

There is a substantial amount of data presented here. This is not the easiest read - partly as it is single space typing and for complex figures like these, it is easier for the reader if they are presented separately- and if there were page numbers. It was also not always easy to understand exactly how the experiment was performed or analyzed as key bits of information were not provided. That said, this is an interesting study - the big question is whether the conclusions are substantiated by the data presented

Major Comments

1: All of these experiments are performed with a single model i.e. 4T1 cells in BALB/c mice. I am sympathetic to researchers being asked to perform the same experiments with different models but, with the focus here on complement factors and recruitment of MDSCs and the known variation in

immune systems between different mouse strains, this is an issue.

2: The most important experiment is shown in Figure 7 and Figure 8.

(a) do the authors have the data shown in Figure 7B-G for the C3aR and C5aR1 antagonists alone - if so this should be shown.

(b) I assume these two figures (plus Supplementary Fig. 5) relate to the same mouse experiment. If so, this should be stated in SFig 5 and Fig. 8 legends. How many mice were in each group? again needs stating. How were the drugs administered?

(c) the data presented in SFig. 5B,C,D should be shown together with the data presented in Figure 8. The supplementary data has the same control group (PBS treated mice) and the observation that the C3aR and C5aR1 antagonists alone do not impact on metastasis is important. But is there a statistically significant decrease in mets in the antagonist alone treated mice versus antagonists with DOX? This is an important comparison. Certainly the PMX 53 alone is as efficient at reducing metastatic incidence as when in combination with DOX.

(e) Fig. 8F - what are the 'a' values on the graph legend - if they related to 'slope' why do these values not match the the median values shown in Fig.8G?

What this comes down to is as follows. In this model, DOX treatment following surgical resection doesn't (as the authors state) alter the number of mice with metastasis, or the number, size or dynamics of the mets. The C3aR and C5aR1 antagonists do reduced metastatic outgrowth when combined with DOX but not the number of mice with mets i.e. no evidence that this combination can eliminate mets/prevent micromets from growing out - the effect is just a modest delay, and as a consequence this limits the impact of the paper.

3: Upregulation of complement factor secretion following DOX treatment.

The authors very convincingly show that the complement factors are predominantly secreted by fibroblasts and back this up with human datasets of lung cancer. What I didn't understand was why there would be higher levels in human breast cancer lung metastases versus bone or brain metastases (SFig. 4C) - the authors conclude a specific role for complement signaling in pulmonary mets - why? there are fibroblasts in other tissues.

Minor comments

Fig.1H-J. Very confusing. Why were some of the mice killed on day 7? Why show the examples IVIS images for day 3 (panel I) when all the mice were alive at Day 7?. In K and L individual points need to be shown.

References should be given for the datasets used

Overall, this a tough read - the figure legends could be much clearer. Although cartoons such as in Fig. 5A are helpful in some ways, it makes it look as if these were non-tumor bearing mice.

Conclusion

I admire the authors for the work that has gone into this manuscript. The questions being asked are highly clinically relevant. The role of complement factors in promoting metastasis via modulation of the niche is increasing. The novelty here is showing the effects of chemotherapy. I am somewhat torn between very much liking the data showing the DOX treatment increased complement factors in the niche and the rather underwhelming data shown in Figure 8 combined with the lack of validation in an independent model.

Reviewer #3 (Remarks to the Author):

Background/summary of the manuscript:

Despite the numerous advances in the development of new targeted therapies and immunotherapy, conventional chemotherapy remains the backbone of systemic treatment of cancer patients after surgery. Indeed, many targeted therapies require current or sequential chemotherapy in order to be effective for upfront induction therapy, e.g. using EGFR or Her2 antibodies. Nevertheless, there are examples in the literature (mainly from preclinical studies) where chemotherapy was shown to actually exacerbate and promote tumor progression, by inducing distant metastatic disease. These pro-tumor effects may reduce or negate the beneficial cytotoxic/anti-proliferative effects of chemotherapy, thereby reducing overall treatment efficacy. One intriguing aspect concerning the efficacy of chemotherapy is when it is used in the adjuvant setting to treat occult, low volume/micrometastatic disease. While there are obviously many examples, based on phase III clinical trials of adjuvant chemotherapy causing prolongation of disease-free survival (DFS) and overall survival (OS), the effects generally tend to be somewhat modest. In some ways this is counterintuitive as one might expect chemotherapy to be more effective than what is generally observed in the adjuvant treatment setting. One possibility for this stems from alterations in the organ or tumor microenvironments such as local tissue damage and/or inflammation which result in an increased ability of tumor cells to grow in such 'damaged' sites and/or to improve tumor cell intravasation and extravasation from tumor blood vessels etc. Such effects have also been observed for other types of therapies, e.g. radiation and antiangiogenic drugs, among others. So chemotherapy may be the tip of an iceberg. Here the authors present a new mechanism to account for chemotherapy-induced pro-metastatic relapse, in this case in breast cancer using a preclinical mouse model (4T1) simulating the triple-negative breast cancer (TNBC) subtype. Basically the authors report the following findings (among others): if normal non-tumor bearing mice are injected with adriamycin (an anthracycline) or cisplatin - and then the mice used as recipients for an intravenous injection of tumor cells, the metastatic burden in the lungs is increased. Second, analysis of lung tissue after chemotherapy of tumor-free mice revealed several changes in isolated fibroblasts, with changes in certain cellular functions and gene expression indicating a pro-inflammatory signature/phenotype. Evidence was also obtained indicating a remodelling of the lung immune microenvironment including an increase in infiltrating granulocytes and eosinophils following systemic chemotherapy of either drug. T cells were found to be increased in the lungs of cisplatin treated mice, but not in the adriamycin-treated mice. Fourth, experiments simulating postsurgical adjuvant therapy indicated that doxorubicin was ineffective in reducing metastatic burden, whereas cisplatin was effective in doing so. Consequently the authors focused most of their remaining experiments on the impact of adriamycin further mechanism related studies, comparing the results to cisplatin treatment.

Some of the main findings from these additional experiments were as follows: adjuvant adriamycin increased in the level of myeloid derived suppressor cells (MDSCs), which contributed to an immunosuppressive microenvironment in the lung metastatic niche. This was secondary to an enhanced infiltration of granulocytes and monocytes, effects that were not observed with cisplatin treatment. Evidence was also found for T cell dysfunction i.e. T cells were found to express markers of immune exhaustion such as LAG3, TIGIT, and PD-1. Next, an interesting series of experiments indicated a role for the above described cellular changes mediated by signaling of the complement system, specifically in lung fibroblasts, involving C3 - a precursor of C3a and C5a, and expression of certain complement receptors (C3aR and C5aR1). Furthermore activation of complement signalling in the lungs was found to be associated with infiltration of C3aR and C5aR1 expressing MDSC's. As a result, the authors assessed the impact of combining adriamycin chemotherapy with blockade of complement signalling for the impact of this combination treatment on preventing lung metastatic/relapse. Complement receptor antagonists such as SB290157 (which blocks C3a) or another antagonist called PMX53 (which blocks C5aR1) were assessed after primary tumor resection in their adjuvant therapy 4T1 breast cancer model. The addition of the inhibitors abrogated the pro-

metastatic effect of adriamycin treatment.

Thus the overall conclusions are that intratumoral fibroblasts can be an ultimate source of inducing an immunosuppressive tumor microenvironment caused by a systemic chemotherapy - at least with adriamycin - and that this is due to expression of certain complement components and complement receptors. Thus it is hypothesized that treatment of TNBC patients, at least, presumably, in the adjuvant setting may benefit from this type of combination treatment.

Critique and comments:

This is a very interesting and detailed study which provides a new perspective on how cancer associated fibroblasts (CAFs) – an area of research that has been the focus of the lab of Dr. Neta Erez for many years - impact tumor growth/progression, metastasis, and response to therapy. The results link several different areas of research, including fibroblast biology, complement signaling, immunotherapy, and metastasis. There is no question of the fact there are a number of intriguing and novel observations. In general, the technical aspects of the work described seem reasonably sound. However, there is, in my opinion, a major flaw in the experimental design, which precludes publication of this manuscript in *Nat Comm.*, at least at this time, but it is a flaw which, if addressed, could elevate the paper to seminal status, depending on the results obtained of some new experiments. Let me explain. The use of single agent adriamycin or cisplatin as a model for postsurgical adjuvant therapy of TNBC is highly questionable. To my knowledge adriamycin or cisplatin are used only as a component of a combination chemotherapeutic adjuvant strategy for TNBC. For example, when adriamycin is used it is often combined with cyclophosphamide, i.e., the so-called “AC” regimen, which is then followed by single agent paclitaxel (“AC→T”). See Burstein: “Patients with triple negative breast cancer: is there an optimal adjuvant treatment?” *The Breast* 2013. An obvious question is whether the prometastatic effects induced by adriamycin in the lungs would not be observed if the drug was combined with another chemotherapeutic agent such as cyclophosphamide or a taxane. Indeed, one might even ask whether cisplatin, if combined with adriamycin, would prevent the prometastatic relapse, based on the observation that cisplatin treatment did not cause the prometastatic effect, and in fact reduced lung metastatic burden. However, since adriamycin combined with cisplatin is not used clinically, as far as I know, it would be preferable to evaluate another drug instead. If such a combination therapy prevented the prometastatic effects caused by adriamycin, this could represent a seminal observation. The reason I say this is that so much of clinical chemotherapy is given as doublet or triplet combinations and the prevailing rationale for using such combinations is that different mechanisms of tumor cell killing, or inhibition cell proliferation, are induced resulting in greater efficacy (albeit at the expense, usually of greater toxicity), and also delaying acquired drug resistance. The present results of this manuscript may indicate an entirely new and novel mechanism for combination therapy, namely, that the prometastatic effects of one drug, in addition to its tumor cell killing ability, may be prevented/blocked by combination with another chemotherapy drug that does not induce the prometastatic effect. I find this possibility to be intriguing and exciting, provided that any new experiments confirm this speculation/hypothesis.

Some other concerns or points/comments include the following:

1) The authors suggest doxorubicin may attenuate anti-cancer immunity and increase lung metastasis through activation of complement signalling of lung fibroblasts in the 4T1 breast cancer model. However, this preclinical phenomenon appears to be somewhat different from clinical results of adjuvant trials. It seems that the extent of lung metastases may vary with the schedule of treatment and endpoint. The authors set different endpoint schedules in the adjuvant setting (Fig.2), early metastasis setting (Fig.3), and the late stage metastasis setting (Fig.8). The increase of lung metastasis numbers in DOX group (compared to PBS group) was only obvious when the endpoint schedule was set earlier in the “early metastasis setting”. In addition, in a previous study reported by Bao L. et al. (*Am J Pathol.* 2011;178(2):838-52.), these authors also used the 4T1 model to test the efficacy of single agent doxorubicin and other chemotherapy regimens. They set the endpoint schedule at least 21 days after the tumor resection and demonstrated that doxorubicin alone or as part of a combinational treatment regimen inhibits lung metastasis compared to the placebo group.

This discrepancy with the authors' current results using the same model reasonably prompts one to suggest a second model for confirmation of lung metastasis between DOX and PBS mice. On the other hand the results do seem to provide a rationale for undertaking postsurgical adjuvant therapy model that was evaluated.

2) In fact, there was a randomized phase 3 trial (PATTERN trial, JAMA Oncology 2020;6(9):1390-1396) conducted in China which compared adjuvant paclitaxel/carboplatin (PCb) and conventional cyclophosphamide / epirubicin / fluorouracil – docetaxel (CEF-T) for TNBC patients. From this trial, we may have a picture of different patient outcomes between platinum-based and anthracycline-based regimens. Although there was no statistical difference in overall survival between groups, patients received platinum-based regimen had a superior DFS (5-year DFS, 86.5%vs 80.3%, hazard ratio = 0.65; 95% CI, 0.44-0.96; P = .03) and less distant metastasis (6.2% vs 10.2%). If we could know the exact patient number of specific metastatic sites especially lung metastasis, then which would provide stronger evidence to support the hypothesis of this study.

3) The authors carried out experiments in a snap shot of a specific time point following chemotherapy administration, usually 24 hours. I wonder for how long these pro-inflammatory effects of fibroblasts are observed? Previous studies in this direction demonstrated that such effects when focusing on immune cells, may hold for up to one week. Is this the case with fibroblasts?

4) A very recent study demonstrated that following PTX therapy, the ECM is altered in the lungs, and as such this could also explain increased metastasis (Haj-Shomaly et al, Cancer Res 2022). This effect was not excluded from the analysis in this study. Especially when focusing on fibroblasts, are there any experiments which have tested ECM remodeling in response to chemotherapy?

5) The authors claim that the CAFs are the main source of cells which contribute to the metastasis. However, some previous studies demonstrated that in addition to fibroblasts, immune cells and tumor cells (actually exosomes) can also promote chemotherapy-induced host responses promoting metastasis. I wonder whether the authors can clearly demonstrate that CAFs blockade or depletion change the metastatic effects in mice in response to chemotherapy.

6) Do the authors know whether the complement receptor antagonists do not affect the viability of cancer cells?

7) Fig. 4e, f showed lungs of doxorubicin-treated mice had increased infiltration of T cells expressing exhaustion/dysfunction markers such as TIM-3, TIGIT, and PD-1. This change in T cells markers might make the immunosuppressive TME niche in lungs more sensitive to immune checkpoint inhibitors such as in response to anti-TIGIT and anti-PD1 antibodies, something that may be worthy of future study of combination therapy involving immune checkpoint inhibitor with doxorubicin, and perhaps certain other drugs.

8) This study suggested that the pro-tumorigenic adverse effects caused by the conventional MTD chemotherapy are largely due to the induction of tissue damage and tumor-promoting inflammation. To alleviate this kind of tissue damage, administering chemotherapy in a so-called "metronomic" low dose way could be a reasonable alternative to consider for future studies. Results of the 'TONIC' trial by Voorwerk et al (Nat Med. 2019 Jun;25(6):920-928) showed that 2 weeks of low-dose doxorubicin treatment on metastatic TNBC, inducing a more favorable immune-active TME with increased T cell infiltration and TCR diversity. Could the dose of doxorubicin used be a major factor causing the different effects on the TME? The dosing used in current study was conventionally used MTD (5mg/kg) which led to the complement-induced immunosuppressive TME niche in lungs. Alizadeh et al (Cancer Res. 2014 Jan 1;74(1):104-18) used the same 4T1 cell line, but in a primary tumor setting. They demonstrated that MTD dosage of doxorubicin (5mg/kg) selectively eliminates MDSC in the spleen, blood, and tumor beds, which seems to be the opposite to what the current study is showing. Could it be that the primary tumors respond differently to doxorubicin therapy compared with the lung metastases in an adjuvant setting?

9) Fig2C and Fig8C clearly show that regarding to prevention of lung metastasis from 4T1 tumor cell in an adjuvant setting, cisplatin alone is much more effective than the combination of doxorubicin and complement signaling inhibitor. If considering more toxicity might be the case with this combination treatment, a switch between doxorubicin and cisplatin should be a much better strategy, especially in a clinical setting – although as mentioned above, use of such drugs as monotherapy regimen is not normally done for adjuvant therapy of TNBC.

10) An obvious concern raised is the limitation of using just one tumor model. How about other mouse TNBC model such as EMT6? This study showed different drugs caused very different TME responses on the same model (4T1); logistically thinking, the same drug (doxorubicin) may have very different effects on different tumor models. I am not suggesting that an additional model be studied replicating all the experiments done in the 4T1 model – but something much more limited such as whether adriamycin or cisplatin have similar enhancing/suppressive effects on lung metastasis, respectively.

11) Finally one final bit of information the authors should know concerns experiments in which tumor free normal mice are treated with a chemotherapy drug, and then tumor cells injected after such treatment, e.g. a day later, and finding that lung metastasis has increased. This is a very interesting result, however, similar experiments were first reported over 40 years ago and then a number of times since then. For example, see Carmel RJ et al, *Cancer Res* 1977 (“The effect of cyclophosphamide and other drugs in the incidence of pulmonary metastases in mice”) and Yamauchi K et al, *Cancer Res* 2008 (“Induction of cancer metastasis by cyclophosphamide pre-treatment of host mice: an opposite effect of chemotherapy”). I think it should also be noted that the literature reporting chemotherapy induction or promotion of metastasis is almost entirely preclinical in nature, and an obvious question is whether there is any definitive clinical trial evidence for chemotherapy promotion of metastasis in patients. By the way the two papers mentioned above both showed that pretreatment of normal mice with cyclophosphamide was a powerful inducer of metastasis. So is this inconsistent with the use of the ‘AC’ regimen in clinical adjuvant therapy of TNBC?

Point-by-point Rebuttal

We would like to thank the reviewers for their overall positive feedback on our study, and for being constructive and thorough. We have addressed all their comments and as a result the revised manuscript is significantly improved.

Reviewer comments

Reviewer #1 (Remarks to the Author):

The manuscript by Monteran et al., describes work showing that doxorubicin, but not cisplatin impacts infiltrating tumor immune cells in the lung and that doxorubicin causes lung fibroblasts to expression complement factors C3aR and C5aR1 that they argue increases metastasis. The observations are interesting and experiments are well controlled. In addition, this is an important area of inquiry.

We thank the reviewer for acknowledging the importance and quality of our study.

An outstanding question, that isn't addressed in the work is whether expression of C3aR and C5aR1 and the associated immune changes are related and more importantly if those changes are responsible for the changes in metastases that is claimed. Of equal importance, it isn't entirely clear to this reviewer if the metastatic changes that are claimed to occur upon treatment with the C3aR or C5aR1 agonists are significant; is there an overall reduction in metastatic tumor burden. These issues should be directly addressed.

We show that the expression of complement components (rather than their receptors- C3aR and C5aR1) are upregulated in fibroblasts following chemotherapy treatment, resulting in chemoattraction of C3aR and C5aR1 expressing immune cells to the metastatic microenvironment. The link of this pathway to metastasis is demonstrated by the fact that treatment with C3aR or C5aR1 antagonists attenuates the observed immune changes, as well as metastatic burden (Fig. 7,8). To better clarify this, we revised Fig. 8b and it now clearly states the effects of treatment with a combination of chemotherapy and C3aR or C5aR1 antagonists on overall lung metastatic incidence and burden. We thank the reviewer for pointing out that this was not sufficiently clear.

Below I provide several points that should be addressed in no particular order of importance.

1. In Figure 1 there are changes reported as a percent of CD45. Do total CD45 cell numbers go up as well or is it only a change in distribution?

We did analyze both numbers and percent of CD45⁺ cell populations. While specific immune cell populations increased in response to therapy (shown in Fig. 1), the total numbers of CD45⁺ cells in lungs were actually slightly decreased in lungs of naïve treated mice, as shown in the graphs below. While we acknowledge the importance of

absolute cell numbers, the ratio of immune cells in a specific microenvironment is a better reflection of the changes in equilibrium between the different functional populations of immune cells. We agree with the reviewer that total numbers may be a better metric when evaluating systemic inflammation for example. However, in the case of the tumor immune microenvironment, the relative changes in the percentage of distinct cell populations is probably a better fit, and is routinely used in similar studies (e.g: Coffelt S. et al, Nature 2015 (PMID: 25822788), Wellenstein et al. Nature 2019, (PMID: 31367040); Klemm et al. Cell 2020, (PMID: 32470396)).

2. The authors suggest that colonization is higher in chemo treated mice compared to PBS treated mice but when one looks at the BLI images they honestly don't look different if one considers the large variation. That being said, there are clear differences at later timepoints. To me that suggests that colonization may not be different but rather survival of seeded cells and/or proliferation is greatly enhanced in the chemo treated mice. Did the authors do BrdU and/or caspase stains early or could they address this point in any other way?

The BLI images shown in Fig. 1 present the earliest detectable time point. The reviewer is correct that at this early time point (3 days) there were no significant differences between the groups (quantified in Fig. 1j). To better clarify this, we added BLI images from day 7, where the differences are clearly apparent, showing that mice that were pre-treated with chemotherapy had enhanced lung colonization. This is now presented in revised Fig. 1i-k.

To address the reviewer's interesting question regarding whether the later significant differences in lung colonization that we observed are related to ability to enter the lungs or are related to differences in survival/growth, we performed additional *in vivo* analyses at early and later time points. To inspect the possible differences in early colonization of tumor cells following pre-conditioning with chemotherapy, we utilized

mCherry-luciferase expressing 4T1 tumor cells that can be detected using IVIS imaging and flow cytometry. At 3h post injection there was a weak luciferase signal that was not different between the groups, suggesting that the injected cells directly entered the lungs following tail vein injection in all groups. However, most of the injected cells likely died, as 24h following injection they were no longer detectable until the 3 days time point, as shown in Fig. 1.

To further address the question regarding survival of tumor cells in lungs of treated vs. non-treated mice, we analyzed cell death by Annexin-V staining to assess whether chemotherapeutic preconditioning affected apoptosis of cancer cells in lungs. Interestingly, there were no significant differences between the groups, as shown below. Thus, pretreatment with chemotherapy does not have a remarkable effect on the initial colonization or on survival of tumor cells. Indeed, we claim that the increased metastatic burden in lungs of chemotherapy-treated mice at later stages results from changes in the lung immune microenvironment, demonstrated in detail throughout our study. To better clarify this point, we revised the term “pulmonary colonization” into “seeding and growth” (Line 111 of the revised manuscript). We thank the reviewer for this suggestion.

(a) Representative IVIS imaging of mice, 3h and 24h post-inoculation with 4T1-mCherry-Luc cells. **(b)** Early lung colonization represented by quantification of luciferase bioluminescence 3h following i.v injection of 4T1 cells. $n= 8, 7,$ and 7 in PBS, DOX, and CIS groups, respectively. Data presented as mean \pm s.e.m **(c-e)** Apoptosis analysis: **(c)** FACS plots of mCherry-expressing cancer cells and Annexin-V at 72h, 7d and 10d following i.v injection of 4T1 cells. $n= 3$ per group in PBS, DOX, and CIS pre-treated mice. **(d,e)** Quantification of apoptosis assay: **(d)** Donut plots representing the average %

of apoptotic cancer cells (mCherry+AnnexinV-FITC+DAPI-), dead cancer cells (mCherry+AnnexinV-FITC-DAPI+) or live cancer cells (mCherry+AnnexinV-FITC-DAPI-) in each group at time points as indicated. (e) % apoptotic cancer cells over time. Data presented as mean \pm s.e.m.

3. In Figure 1, the authors argue that doxorubicin and cisplatin induced similar inflammatory/fibrosis signatures and in the seeding assay, both increased tumor outgrowth. In Figure 2 when examining spontaneous metastasis following resection, cisplatin reduces metastasis while doxorubicin does not. What is the disconnect in the two assays? Could this be a dosing effect, in other words was the maximal tolerated dose used in both chemotherapies (i.e., I know that 5 mg/kg is not the maximum dose for doxorubicin but I do not know if that is true for cisplatin)? This seems to be an important observation that should be discussed and a dose course might be helpful here.

We thank the reviewer for raising these points. Indeed, one can find reports in literature using doses higher than 5mg/kg. However, we carefully calibrated the doses in our models and since higher concentrations were lethal to all mice, the doses used in our manuscript are the MTD in our system.

Regarding the difference between the two chemotherapies shown in Figure 1 and 2: in Figure 1 we treated naïve/healthy mice with chemotherapy whereas in Figure 2, mice received two cycles of chemotherapy after they already had a primary tumor grown and surgically resected. These are very different physiological conditions. While in Figure 1 chemotherapy induced tissue damage and pro-inflammatory signaling, in Figure 2 the effects of chemotherapy also included tumor cell killing. Thus, the effect of cisplatin described in Figure 1 is limited to the normal host tissue response whereas in Figure 2 the pro-inflammatory effects may be counteracted by the direct killing of cancer cells by chemotherapy.

We agree with the reviewer that the differences in efficacy between Dox and Cisplatin are very interesting. We therefore expanded our investigation of the differences they induced in the immune milieu in treated mice. These new findings and analysis are included throughout the revised manuscript in Fig. 3, Supp. Fig.3, Supp. Fig. 4, Supp. Fig. 6, Supp. Fig. 8.

4. Figure 2D shows metastatic area, is there a difference between PSB and doxorubicin? Based on the argument put forward, there should be greater metastasis in doxorubicin but no p values is provided. This needs to be explained/addressed.

As shown in Figure 2, there were no significant differences (p values are shown for significant differences) in the metastatic burden of doxorubicin-treated mice compared with control/PBS treated mice in adjuvant setting following surgical tumor removal. While cisplatin significantly reduced metastasis, doxorubicin had no therapeutic effect in inhibiting metastasis compared to PBS. In other words: doxorubicin was not efficient in this setting of end-stage in reducing metastasis, which motivated us to further study the ineffectiveness of doxorubicin.

5. Figure 3B shows increased metastasis 24 hours after doxorubicin compared to PBS, but that does not equate to increased metastasis (Figure 2) at what looks to be one week after the last doxorubicin (Figure 2A but the day isn't listed on the figure, it should be). What does this early increase mean if it does not equate to increased metastasis?

We agree with the reviewer regarding this conundrum. This discrepancy may be explained by the fact that the overall effect of chemotherapy is the sum of its beneficial effects (killing of cancer cells) and its deleterious adverse effects including the induction of tissue damage and inflammation. Our findings suggest that at early stages, the tissue damage and tumor-promoting inflammation induced by doxorubicin counteract the anti-cancer effects and allow for more metastases formation compared with control, while at later stages, the balance of these two contradicting effects resulted in no therapeutic benefit. This prompted us to investigate these adverse effects and overcome them by co-targeting of chemotherapy-induced inflammatory pathways. To address this comment, we discussed this more clearly in the revised manuscript (results section; lines 148-154).

As for indicating the days in Figure 2a, we have corrected this in the revised version and thank the reviewer for pointing this out.

6. Figure 3 shows fold changes in immune populations, it would be more informative to show percent CD45 (main figure) and total cell counts (in supplementary) to provide more insight into what these changes really look like.

The results are presented as fold change of the average percent CD45 from several combined different biological repeats. Moreover, the donut plots in Figure 3c indicate the average percent of CD45 cells in the different groups, providing further insights on the distribution of each cell type. To address the reviewer's comment, we added cell numbers in new Supplementary Fig. 3j of the revised manuscript, to show the specific changes, as requested.

7. Figure 4 addresses the functional state of myeloid cells following chemotherapy and so compares PBS versus doxorubicin treated mice. The setup was that doxorubicin is different than cisplatin so it would be nice to show there are no suppressor, or reduced suppressor cells in cisplatin lungs immediately after treatment (again, this after showing both chemotherapies are at their maximal doses).

The analysis in Figure 3 indicates that in doxorubicin-treated mice there was an increase in infiltration of myeloid cells, whereas their infiltration in cisplatin-treated mice was not different than control mice. This prompted us to focus on the myeloid cells recruited to dox-treated lungs in Figure 4, and show that they are immunosuppressive. To address the reviewer's question, we now analyzed gene expression in the recruited immune cells in lungs of cisplatin-treated mice and found that they were significantly less immunosuppressive. This new analysis is presented

in revised Supp. Fig 6a-c), providing further mechanistic insights on the differences between Dox and cisplatin. Thus, doxorubicin treatment induces enhanced recruitment of myeloid cells as well as elevated immunosuppressive gene expression. We thank the reviewer for this suggestion.

8. Figure 4F and beyond, as stated earlier, it would be more informative to shown percent of CD45 and total numbers opposed to fold changes, are significance values maintained?

As detailed for similar points above, we did analyze both numbers and percent of CD45⁺ cell populations. While there were no significant differences in total numbers, there were significant changes in the abundance of specific immune cell populations increased in response to therapy, leading to skewed function and reshaping of the immune metastatic niche towards resistance.

9. Figure 4G, T cell suppression assays are typically presented with dilutions to truly assess the suppressive nature and a control with no “suppressor” added to assess maximal T cell proliferative capacity in the assay. These should be added to the figure.

We agree with the reviewer that T cell suppression assays frequently include several dilutions of the suppressor factor/cell. However, in our case, the suppressive activity originates from MDSCs that we have isolated by FACS from metastases-bearing mice. This is a non-trivial challenge that does not allow such dilutions with an ethically acceptable number of mice. Therefore, we calibrated the functional effect and performed the experiment with the selected number of MDSCs, in 4 biological repeats. To address the concern regarding the “no suppressor” control, we added the relevant control of T cell proliferation in the absence of any MDSCs. The results are shown in revised Figure 4h. We thank the reviewer for pointing this out.

10. Gating strategies for all flow should be provided in the supplementary data so the reviewer/reader can evaluate the FACS data. Figure 4J is looking at checkpoint expression and there isn't an obvious population, just a shift, which is not a surprise per se but evaluation of the gating would provide more confidence in the differences being reported.

We thank the reviewer for pointing this out. We added supplementary figures with gating strategies for all FACS experiments (revised Supp. Fig. 2a,b and Supp. Fig. 6d).

11. Suppl Figure 3, the PCA plot shows that one of the doxorubicin treated mice is quite distinct from the other two, could the authors discuss the variation they observe?

Indeed, one of the dox-treated mice was different than the other two. However, it was still very distinct from the PBS-treated mice and more similar to the other two dox-treated mice, as seen in revised Supp. Figure 6h,i. Ideally, we would be able to connect this variability to clinical outcome, but in this case it is not possible as the mice were

killed at an early point following treatment. Notably, variability is inherent to *in vivo* experiments and while indeed the dox group showed variability, it was still clear that treatment with dox profoundly altered the gene expression of lung fibroblasts.

12. They state that they are comparing their single cell murine data to human metastatic single cell data (GSE14020) in Supple Figure 4. This is array data of metastasis and this should be clarified as much of the signal could (or may not) come from tumor cells.

We thank the reviewer for bringing to our attention that this was not sufficiently clear. We clarified this in the revised legends of Supp. Fig.7.

13. They have compared and contrasted cisplatin to doxorubicin. In Figure 5, if cancer cells are treated with cisplatin and that is used on lung fibroblasts, what happens to complement expression? This seems to be an important control.

To address this interesting point, we tested the effect of cisplatin treatment on complement expression in fibroblasts. Strikingly, in contrast to doxorubicin, treatment of lung fibroblasts with CM of cisplatin-treated 4T1 did not induce upregulation of complement factors. These data are now shown in revised Supplementary Fig. 8b,c). Moreover, to further address the reviewer's question, we analyzed C3 secretion and complement score *in vivo* and found that they were not elevated following cisplatin treatment, compared with dox and PBS in two different models: 4T1 and EO771. The results are shown in new Supp. Fig. 8f. Thus, the fact that cisplatin was more efficient in reducing lung metastasis may be at least partially due to its inability to induce complement signaling in fibroblasts. We thank the reviewer for this suggestion.

14. Only the C3aR antagonist reduced immune infiltration while both the C3aR and C5aR1 antagonist reduced metastasis in Figure 8, does this suggest it is not the immune infiltrate that is responsible for the differences in metastasis? This seems contrary to their hypothesis.

C3aR and C5aR antagonists had differential effects on immune cells. While C3aR antagonist affected multiple populations of myeloid cells, C5aR antagonist reduced the infiltration and numbers of dysfunctional T cells, as seen in Figure 7. This may explain their differential mechanisms in attenuating metastatic relapse. However, via their effect on immune cell composition and function, both antagonists significantly attenuated metastasis. This is now presented more clearly in revised Fig. 8b.

15. In Figure 8 they argue that the number of metastases are reduced when a C3aR or C5aR1 agonist is combined with doxorubicin but what happens to overall burden, is it reduced, if it is not, this is not that impressive.

To better clarify this, we revised Fig. 8b and it now clearly shows the significant effects of treatment with a combination of chemotherapy and C3aR or C5aR1 antagonists on

overall lung metastatic incidence and burden. We thank the reviewer for pointing out that this was not sufficiently clear.

16. In regard to Figure 8, it is important to show the tumor burdens when mice are treated with C3aR or C5aR1 agonist alone on the same graphs --- currently these data are in the supplemental and it looks like the C5aR1 agonist may change lung burden and C3aR agonist may impact lung number but is it impossible to tell with them on different graphs and the axes are different.

To address these points, and also in response to comments from other reviewers, we revised the analysis of metastasis in Figure 8. We added the data of C3aR and C5aR antagonists alone, and included analysis of overall metastatic incidence in all the groups (% mice with no mets, unifocal or multifocal mets). This is now shown in new Figure 8b (metastatic burden, overall metastatic incidence) and in new Supp. Figure 9 (# of lung mets, metastatic area). We also corrected the different axes to be uniform in revised Supp. Figure 9, and in the revised manuscript text, as suggested. We thank the reviewer for bringing this to our attention.

17. Also Figure 8, I know what a slope is, but it would help if the authors described how this was calculated for these data.

The slope was calculated using two time points representing the number of visible lung metastases visualized in CT scans. To address this point, we better clarified this in the revised manuscript text (Lines 362-366 and in Fig. 8 legend).

Reviewer #2 (Remarks to the Author):

In this manuscript Monteran and colleagues have addressed a highly clinically relevant question - namely what are the adverse effects of chemotherapy on normal tissue damage, how this impacts metastatic colonization and are there approaches that can be taken to counteract the adverse effects of chemotherapy whilst retaining its anti-tumor effects.

We thank the reviewer for acknowledging the importance and clinical relevance of our study.

The authors have used the 4T1 model of breast cancer metastasis in syngeneic BALB/c mice. They clearly demonstrate that anthracycline chemotherapy (doxorubicin or cisplatin) clearly impact the normal lung tissue (changes in fibroblasts gene expression, changes in immune cell populations) - observations that in themselves are not novel but set the scene for their subsequent studies. What is interesting is that the different chemotherapies have different effects and the authors provide evidence (see comments below) for increased metastatic outgrowth in mice pretreated with chemo and then inoculated intravenously with 4T1 cells. For the majority of subsequent

experiments, the authors predominantly use a model whereby 4T1 primary tumors are resected and then the mice treatment with chemo - this is to be commended as it mimics the treatment scenario for the majority of breast cancer patients (although increasingly patients with TNBC are treated with neo-adjuvant chemotherapy). Again the authors show changes in the immune compartments following early and later stage metastatic colonization, and in the transcription profile of isolated lung fibroblasts, particularly of the upregulated expression of complement genes. Finally, the key experiment, the authors demonstrate that compared to mice treated with doxorubicin alone, combination treated with complement receptor antagonists reduces metastatic outgrowth in the lungs.

There is a substantial amount of data presented here. This is not the easiest read - partly as it is single space typing and for complex figures like these, it is easier for the reader if they are presented separately- and if there were page numbers. It was also not always easy to understand exactly how the experiment was performed or analyzed as key bits of information were not provided.

We apologize that the reviewer found the manuscript difficult to read. In the revised manuscript, we tried to better clarify how experiments were performed and included page numbers and larger spaces.

That said, this is an interesting study - the big question is whether the conclusions are substantiated by the data presented

Major Comments

1. All of these experiments are performed with a single model i.e. 4T1 cells in BALB/c mice. I am sympathetic to researchers being asked to perform the same experiments with different models but, with the focus here on complement factors and recruitment of MDSCs and the known variation in immune systems between different mouse strains, this is an issue.

We agree with the reviewer on the importance of validating biological findings in more than one model. To address this concern, we calibrated and performed the relevant adjuvant treatment experiments in an additional model of triple negative breast cancer- the EO771 transplantable model (C57BL/6 background). Strikingly, these additional sets of experiments yielded similar results to our findings in the 4T1-based model (BALB/c background). The new results were integrated into multiple figures in the revised manuscript (Fig. 2f-h, Fig. 3d-f, Supp. Fig. 3, Supp. Fig 6e-g, Supp. Fig. 8d-f). Specifically, analysis of pulmonary metastatic load at late-stage showed that doxorubicin did not impact metastases formation as compared to control, while cisplatin treatment resulted in a trend towards lower metastatic load as compared to control ($p=0.07$) (New Fig. 2f-h). We also analyzed the lung immune microenvironment at an early metastatic stage and found changes in lung infiltrating granulocytes and monocytes that were similar to what we found in 4T1 injected mice (Fig. 3d-f, Supp.

Fig. 3). Moreover, the lungs of the doxorubicin-treated group had higher complement scores compared with the control group in both models (Fig. 6i, Supp. Fig. 8f). Most impressively, C3a protein levels were significantly elevated in lungs of doxorubicin-treated mice as compared to lungs derived from either cisplatin-treated mice or from control mice (Supp. Fig. 8d,e). Finally, we show that doxorubicin treatment led to increased accumulation of dysfunctional T cells in lungs of EO771-injected mice (Supp. Fig. 6e-g).

While these experiments were intense, we want to thank the reviewer for this important suggestion which significantly strengthened the impact of our findings.

2. The most important experiment is shown in Figure 7 and Figure 8.

(a) do the authors have the data shown in Figure 7B-G for the C3aR and C5aR1 antagonists alone - if so this should be shown.

The distribution of immune populations following treatment with C3aR and C5aR1 antagonists alone is now shown in donut plots in revised Supp. Fig. 9b. To further address the reviewer's comment, we also added new graphs in revised Supp. Fig. 9a-g showing the effect of antagonists alone on complement score as well as on immune-cell populations, similar to what is shown in Fig 7.

(b) I assume these two figures (plus Supplementary Fig. 5) relate to the same mouse experiment. If so, this should be stated in SFig 5 and Fig. 8 legends. How many mice were in each group? again needs stating. How were the drugs administered?

We thank the reviewer for bringing to our attention that this was not sufficiently clear. The early (Fig. 7) and late (Fig. 8) time point experiments were performed separately, each with multiple repeats and the controls for each set of experiments are shown in the corresponding supplementary figures (new Supp. Fig. 9). Specifically, the early time point analysis (Fig. 7) including treatment with antagonists alone (Supp. Fig. 9) was performed three times, and the analysis of end-point metastasis (Fig. 8) was performed 2-3 times. Drugs were administered i.p. We now stated this information more clearly in the revised manuscript and indicated the number of mice analyzed.

(c) the data presented in SFig. 5B,C,D should be shown together with the data presented in Figure 8. The supplementary data has the same control group (PBS treated mice) and the observation that the C3aR and C5aR1 antagonists alone do not impact on metastasis is important. But is there a statistically significant decrease in mets in the antagonist alone treated mice versus antagonists with DOX? This is an important comparison. Certainly the PMX 53 alone is as efficient at reducing metastatic incidence as when in combination with DOX.

To address these important points, and also in response to similar comments from other reviewers, we revised the analysis of metastasis in Figure 8. We added the data of C3aR and C5aR antagonists alone, and included analysis of overall metastatic

incidence (% mice with no mets, unifocal or multifocal mets- new Figure 8b, new Supp. Figure 9j-l), metastatic burden (sum of metastatic area/lung) and # of lung metastases. Regarding the effect of PMX53 alone, indeed there is no statistical difference in the effect on metastasis compared with PMX53+dox. However, statistical analysis of all the comparisons showed that the combination of PMX53 with DOX significantly reduced metastasis when compared to control (PBS) or to Dox alone, while PMX53 alone did not significantly affect overall metastatic incidence and burden. Moreover, although not statistically significant, combining PMX53 with DOX reduced incidence of multifocal metastases compared to PMX53 alone. This is now shown more clearly in new Fig. 8b and in new Supp. Fig. 9. We thank the reviewer for this comment.

(d) Fig. 8F - what are the 'a' values on the graph legend - if they related to 'slope' why do these values not match the the median values shown in Fig.8G?

The 'a' values are indeed the slope values, and they represent the average value of the slope from each treatment group. We have better clarified this in the revised legend of Fig. 8. As for their relation to data shown in new Fig. 8f, the violin plots show the slope of individual mice, and the average (the 'a' values, which match the "a" values shown in Fig. 8e) is shown as a dashed line. To make this clearer, we changed the dashed line in revised Figure 8f to red, and stated this clearly in the figure legends. We thank the reviewer for pointing this out.

What this comes down to is as follows. In this model, DOX treatment following surgical resection doesn't (as the authors state) alter the number of mice with metastasis, or the number, size or dynamics of the mets. The C3aR and C5aR1 antagonists do reduced metastatic outgrowth when combined with DOX but not the number of mice with mets i.e. no evidence that this combination can eliminate mets/prevent micromets from growing out - the effect is just a modest delay, and as a consequence this limits the impact of the paper.

We agree with the reviewer that the combined treatment did not completely cure the mice. However, the C3aR and C5aR1 antagonists did actually reduce overall metastatic incidence and burden. To better elucidate this, we revised Fig. 8b to better represent the results. Moreover, we showed that treatment with doxorubicin accelerated the timeline of metastatic relapse (Figure 3b, and higher slope shown in Figure 8e,f). The fact that the slope of metastatic progression in the combined treatment is clearly attenuated (Fig. 8e,f) indicates inhibition of the rate of progression and resulted in more mice that had only one metastatic lesion, rather than an advanced multifocal disease. We believe that in clinical terms these are not modest effects, and could be translated to attenuation of metastatic relapse and slower progression in human patients treated following resection of their primary tumor. Thus, our data provide a novel strategy that can improve existing treatments and postpone the progression to advanced metastatic disease. We better explained this in the revised manuscript text (Line 345-354 and 362-366).

3. Upregulation of complement factor secretion following DOX treatment.

The authors very convincingly show that the complement factors are predominantly secreted by fibroblasts and back this up with human datasets of lung cancer.

We thank the reviewer for acknowledging this main point of our study- that CAFs are a main source of complement signaling in lungs.

What I didn't understand was why there would be higher levels in human breast cancer lung metastases versus bone or brain metastases (SFig. 4C) - the authors conclude a specific role for complement signaling in pulmonary mets - why? there are fibroblasts in other tissues.

This is an interesting point that we can only speculate on. One possible explanation of the difference between bone, brain and lung metastasis is the fact that there are stromal cells other than fibroblasts in the brain and bone (astrocytes, osteoblasts) that have very different functional roles. Moreover, our hypothesis is that complement secretion by fibroblasts is part of their response to DAMPs (as we have previously shown- Ershaid et al. Nature Comm. 2019). Since lungs are an organ that is exposed to the exterior, they are rich with mucosal immunity and their stromal cells participate in tissue protective pathways. Following this line of thought, one could hypothesize that fibroblasts in the GI tract may be similar to lung fibroblasts in this context. Indeed, we have preliminary data showing that mouse colon fibroblasts are also a main source of complement factors following tissue damage. This is a separate future study that we are currently following up on, which we hope to be able to share in the future. We thank the reviewer for this intriguing thought.

Minor comments

Fig.1H-J. Very confusing. Why were some of the mice killed on day 7? Why show the examples IVIS images for day 3 (panel I) when all the mice were alive at Day 7?. In K and L individual points need to be shown.

We are sorry that the reviewer found these data confusing. Mice were killed at humane time points, when they had visible clinical signs of advanced disease. The dynamics of disease progression in mice treated with cisplatin were dramatically faster. In mice treated with dox, some mice died on day 7 and others survived longer, as shown in Fig. 1j. To better clarify this point, we performed the experiment again, added IVIS images from day 7, and quantified the luciferase signal following both pre-treatments (dox and cis), showing that although the dynamics were different between the treatments, they both resulted in significant elevation of lung colonization. These results are shown in revised Fig. 1i-k.

References should be given for the datasets used.

The references for the datasets used are provided in the figure legends of Fig. 5e, Supp. Fig. 7. They are also referenced in the manuscript text.

Overall, this a tough read - the figure legends could be much clearer. Although cartoons such as in Fig. 5A are helpful in some ways, it makes it look as if these were non-tumor bearing mice.

We revised and improved the clarity of the figure legends throughout the revised manuscript. We also corrected the cartoon in Fig. 5a to clarify that these were mice at early metastatic stage.

Conclusion

I admire the authors for the work that has gone into this manuscript. The questions being asked are highly clinically relevant. The role of complement factors in promoting metastasis via modulation of the niche is increasing. The novelty here is showing the effects of chemotherapy. I am somewhat torn between very much liking the data showing the DOX treatment increased complement factors in the niche and the rather underwhelming data shown in Figure 8 combined with the lack of validation in an independent model.

We are thankful to the reviewer for their overall very positive and constructive assessment of our work and its clinical importance. We hope that now that we added data from another model of TNBC the impact and novelty of our findings are even stronger. We truly appreciate the time and thought put into this review and feel that the comments helped us improve our study.

Reviewer #3 (Remarks to the Author):

Background/summary of the manuscript:

Despite the numerous advances in the development of new targeted therapies and immunotherapy, conventional chemotherapy remains the backbone of systemic treatment of cancer patients after surgery. Indeed, many targeted therapies require current or sequential chemotherapy in order to be effective for upfront induction therapy, e.g. using EGFR or Her2 antibodies. Nevertheless, there are examples in the literature (mainly from preclinical studies) where chemotherapy was shown to actually exacerbate and promote tumor progression, by inducing distant metastatic disease. These pro-tumor effects may reduce or negate the beneficial cytotoxic/anti-proliferative effects of chemotherapy, thereby reducing overall treatment efficacy. One intriguing aspect concerning the efficacy of chemotherapy is when it is used in the adjuvant setting to treat occult, low volume/micrometastatic disease. While there are obviously many examples, based on phase III clinical trials of adjuvant chemotherapy causing prolongation of disease-free survival (DFS) and overall survival (OS), the

effects generally tend to be somewhat modest. In some ways this is counterintuitive as one might expect chemotherapy to be more effective than what is generally observed in the adjuvant treatment setting. One possibility for this stems from alterations in the organ or tumor microenvironments such as local tissue damage and/or inflammation which result in an increased ability of tumor cells to grow in such 'damaged' sites and/or to improve tumor cell intravasation and extravasation from tumor blood vessels etc. Such effects have also been observed for other types of therapies, e.g. radiation and antiangiogenic drugs, among others. So chemotherapy may be the tip of an iceberg. Here the authors present a new mechanism to account for chemotherapy-induced pro-metastatic relapse, in this case in breast cancer using a preclinical mouse model (4T1) simulating the triple-negative breast cancer (TNBC) subtype. Basically the authors report the following findings (among others): if normal non-tumor bearing mice are injected with adriamycin (an anthracycline) or cisplatin - and then the mice used as recipients for an intravenous injection of tumor cells, the metastatic burden in the lungs is increased. Second, analysis of lung tissue after chemotherapy of tumor-free mice revealed several changes in isolated fibroblasts, with changes in certain cellular functions and gene expression indicating a pro-inflammatory signature/phenotype. Evidence was also obtained indicating a remodelling of the lung immune microenvironment including an increase in infiltrating granulocytes and eosinophils following systemic chemotherapy of either drug. T cells were found to be increased in the lungs of cisplatin treated mice, but not in the adriamycin-treated mice. Fourth, experiments simulating postsurgical adjuvant therapy indicated that doxorubicin was ineffective in reducing metastatic burden, whereas cisplatin was effective in doing so. Consequently the authors focused most of their remaining experiments on the impact of adriamycin further mechanism related studies, comparing the results to cisplatin treatment.

Some of the main findings from these additional experiments were as follows: adjuvant adriamycin increased in the level of myeloid derived suppressor cells (MDSCs), which contributed to an immunosuppressive microenvironment in the lung metastatic niche. This was secondary to an enhanced infiltration of granulocytes and monocytes, effects that were not observed with cisplatin treatment. Evidence was also found for T cell dysfunction i.e. T cells were found to express markers of immune exhaustion such as LAG3, TIGIT, and PD-1. Next, an interesting series of experiments indicated a role for the above described cellular changes mediated by signaling of the complement system, specifically in lung fibroblasts, involving C3 - a precursor of C3a and C5a, and expression of certain complement receptors (C3aR and C5aR1). Furthermore activation of complement signalling in the lungs was found to be associated with infiltration of C3aR and C5aR1 expressing MDSC's. As a result, the authors assessed the impact of combining adriamycin chemotherapy with blockade of complement signalling for the impact of this combination treatment on preventing lung metastatic/relapse. Complement receptor antagonists such as SB290157 (which blocks C3a) or another antagonist called PMX53 (which blocks C5aR1) were assessed after primary tumor resection in their adjuvant therapy 4T1 breast cancer

model. The addition of the inhibitors abrogated the pro-metastatic effect of adriamycin treatment.

Thus the overall conclusions are that intratumoral fibroblasts can be an ultimate source of inducing an immunosuppressive tumor microenvironment caused by a systemic chemotherapy - at least with adriamycin - and that this is due to expression of certain complement components and complement receptors. Thus it is hypothesized that treatment of TNBC patients, at least, presumably, in the adjuvant setting may benefit from this type of combination treatment.

Critique and comments:

This is a very interesting and detailed study which provides a new perspective on how cancer associated fibroblasts (CAFs) – an area of research that has been the focus of the lab of Dr. Neta Erez for many years - impact tumor growth/progression, metastasis, and response to therapy. The results link several different areas of research, including fibroblast biology, complement signaling, immunotherapy, and metastasis. There is no question of the fact there are a number of intriguing and novel observations. In general, the technical aspects of the work described seem reasonably sound.

We thank the reviewer for the excellent summary of our main findings and for acknowledging the novelty, quality and importance of our study in linking several areas of research and in providing new perspectives.

However, there is, in my opinion, a major flaw in the experimental design, which precludes publication of this manuscript in Nat Comm., at least at this time, but it is a flaw which, if addressed, could elevate the paper to seminal status, depending on the results obtained of some new experiments. Let me explain. The use of single agent adriamycin or cisplatin as a model for postsurgical adjuvant therapy of TNBC is highly questionable. To my knowledge adriamycin or cisplatin are used only as a component of a combination chemotherapeutic adjuvant strategy for TNBC. For example, when adriamycin is used it is often combined with cyclophosphamide, i.e., the so-called "AC" regimen, which is then followed by single agent paclitaxel ("AC→T"). See Burstein: "Patients with triple negative breast cancer: is there an optimal adjuvant treatment?" The Breast 2013. An obvious question is whether the prometastatic effects induced by adriamycin in the lungs would not be observed if the drug was combined with another chemotherapeutic agent such as cyclophosphamide or a taxane.

Indeed, one might even ask whether cisplatin, if combined with adriamycin, would prevent the prometastatic relapse, based on the observation that cisplatin treatment did not cause the prometastatic effect, and in fact reduced lung metastatic burden. However, since adriamycin combined with cisplatin is not used clinically, as far as I know, it would be preferable to evaluate another drug instead. If such a combination therapy prevented the prometastatic effects caused by adriamycin, this could represent a seminal observation. The reason I say this is that so much of clinical chemotherapy is given as doublet or triplet combinations and the prevailing rationale for using such combinations is that different mechanisms of tumor cell killing, or inhibition cell proliferation, are induced resulting in greater efficacy (albeit at the expense, usually of greater toxicity), and also delaying acquired drug resistance. The present results of this manuscript may indicate an entirely new and novel mechanism for combination therapy, namely, that the prometastatic effects of one drug, in addition to its tumor cell killing ability, may be prevented/blocked by combination with another chemotherapy drug that does not induce the prometastatic effect. I find this possibility to be intriguing and exciting, provided that any new experiments confirm this speculation/hypothesis.

To address the reviewer's concern regarding the possibility that combination treatment of doxorubicin with cyclophosphamide could abrogate the adverse effects following doxorubicin treatment, we performed new *in vivo* experiments. Mice were treated with a combination of doxorubicin and cyclophosphamide following resection of their primary tumor. We found similar results to treatment with doxorubicin alone: AC treatment resulted in a significant increase in recruitment of granulocytes, monocytes and T cells. Moreover, while the combination treatment (DOX+CTX) attenuated the progression rate, the metastatic burden at end-stage was not different than dox treatment or no treatment. These results are presented in new Supplementary Fig. 5. Thus, the combination treatment, commonly used in TNBC patients has similar adverse effects.

We would like to point out that while we agree with the reviewer that combination chemotherapy is the standard of care for early TNBC in an adjuvant setting, single agent chemotherapy based on anthracyclines including doxorubicin is often used to treat advanced/metastatic TNBC. For example: PMID: 18421049

In the current study, we aimed to elucidate the paradoxical adverse effects of chemotherapeutic agents used to treat TNBC. Therefore, we investigated the effects of chemotherapeutic agents as single-agent regimens and not in combinatory regimens that might preclude drawing conclusions regarding specific agents. Moreover, while the mechanisms of cyclophosphamide as a single-agent has been extensively studied, including its effects on the pre-metastatic microenvironment, the effects of doxorubicin and cisplatin on shaping the pre-metastatic microenvironment are still largely unknown. Further, to our knowledge, there are no reports on the effect of adjuvant chemotherapy on the pre-metastatic microenvironment using a

spontaneous model of resectable TNBC, as most studies inspect chemotherapy-induced damage in a primary tumor setting, which is less clinically relevant to metastatic relapse. Therefore, we focused our investigations on deciphering these mechanisms. We thank the reviewer for raising this interesting point.

Some other concerns or points/comments include the following:

1. The authors suggest doxorubicin may attenuate anti-cancer immunity and increase lung metastasis through activation of compliment signalling of lung fibroblasts in the 4T1 breast cancer model. However, this preclinical phenomenon appears to be somewhat different from clinical results of adjuvant trials. It seems that the extent of lung metastases may vary with the schedule of treatment and endpoint. The authors set different endpoint schedules in the adjuvant setting (Fig.2), early metastasis setting (Fig.3), and the late stage metastasis setting (Fig.8). The increase of lung metastasis numbers in DOX group (compared to PBS group) was only obvious when the endpoint schedule was set earlier in the “early metastasis setting”.

The treatment regimen in all our pre-clinical experiments was the same: two doses of chemotherapy were administered 4 and 9 days following surgical removal of the primary tumor. We analyzed the changes in the metastatic niche at two defined time points: early metastatic stage (24h after the second dose of chemo, Fig. 3), to assess the chemotherapy-induced changes in fibroblasts and immune cells, and late metastatic stage (~2-3 weeks following tumor resection), to assess the effect on metastases formation (Fig 2,8). These time points were determined by careful longitudinal calibrations with CT imaging to determine the distinct timing of early and late stages. We better clarify this in the revised figure legends.

As for the counterintuitive observations whereby there was an increase in micromets in dox-treated mice (Fig. 3b) but no benefit and no significant different to PBS at end-stage (Fig. 2). This discrepancy may be explained by the fact that the overall effect of chemotherapy is the sum of its beneficial effects (killing of cancer cells) and its deleterious adverse effects including the induction of tissue damage and inflammation. Our findings suggest that at early stages, the tissue damage and tumor-promoting inflammation induced by doxorubicin counteract the anti-cancer effects and allow for more metastases formation compared with control, while at later stages, the balance of these two contradicting effects resulted in no therapeutic benefit. This prompted us to investigate these adverse effects and overcome them by co-targeting of chemotherapy-induced inflammatory pathways. To address this comment, we discussed this more clearly in the revised manuscript (results section; lines 148-154).

In addition, in a previous study reported by Bao L. et al. (Am J Pathol. 2011;178(2):838-52.), these authors also used the 4T1 model to test the efficacy of single agent doxorubicin and other chemotherapy regimens. They set the endpoint schedule at least 21 days after the tumor resection and demonstrated that doxorubicin alone or as part of a combinational treatment regimen inhibits lung metastasis

compared to the placebo group. This discrepancy with the authors' current results using the same model reasonably prompts one to suggest a second model for confirmation of lung metastasis between DOX and PBS mice. On the other hand the results do seem to provide a rationale for undertaking postsurgical adjuvant therapy model that was evaluated.

In the study by BAO et al. the authors treated mice with 5mg/kg dox continuously every other day for 21 days, which amounts to approximately 50mg/kg cumulative dose. In our model and in most of the recent literature, the maximum tolerated cumulative dose is far lower. We agree with the reviewer that an additional model would strengthen the impact of our findings. Therefore, also in response to a comment by another reviewer, we performed the same experiments with another model of TNBC, the EO771 model, in C57BL/6 mice. Strikingly, this additional set of experiments yielded similar results to our findings in the 4T1-based model (BALB/c background). The new results were integrated into multiple figures in the revised manuscript (Fig. 2f-h, Fig. 3d-f, Supp. Fig. 3, Supp. Fig 6e-g, Supp. Fig. 8d-f). Specifically, analysis of pulmonary metastatic load at late-stage showed that doxorubicin did not impact metastases formation as compared to control, while cisplatin treatment resulted in a trend towards lower metastatic load as compared to control ($p=0.07$) (new Fig. 2f-h). We also analyzed the lung immune microenvironment at an early metastatic stage and observed changes in lung infiltrating granulocytes and monocytes that were similar to what we found in 4T1 injected mice (Fig. 3d-f). Moreover, lungs of the doxorubicin-treated group had higher complement scores compared to control group (Supplementary Fig. 8f). Most impressively, C3a protein levels were significantly elevated in lungs of doxorubicin-treated mice as compared to lungs derived from either cisplatin-treated mice or from control mice (Supp. Fig. 8d,e). Finally, we show that doxorubicin treatment led to increased accumulation of dysfunctional T cells in lungs of EO771-injected mice (Supp. Fig. 6d,e). We thank the reviewer for this suggestion.

2. In fact, there was a randomized phase 3 trial (PATTERN trial, JAMA Oncology 2020;6(9):1390-1396) conducted in China which compared adjuvant paclitaxel/carboplatin (PCb) and conventional cyclophosphamide / epirubicin / fluorouracil – docetaxel (CEF-T) for TNBC patients. From this trial, we may have a picture of different patient outcomes between platinum-based and anthracycline-based regimens. Although there was no statistical difference in overall survival between groups, patients received platinum-based regimen had a superior DFS (5-year DFS, 86.5%vs 80.3%, hazard ratio = 0.65; 95% CI, 0.44-0.96; $P = .03$) and less distant metastasis (6.2% vs 10.2%). If we could know the exact patient number of specific metastatic sites especially lung metastasis, then which would provide stronger evidence to support the hypothesis of this study.

We thank the reviewer for this excellent point and for providing us the information regarding this clinical trial. We tried to reach out to the authors of the trial, but sadly

we did not get any response. We nevertheless included this in the discussion of the revised manuscript line 415-421.

3. The authors carried out experiments in a snapshot of a specific time point following chemotherapy administration, usually 24 hours. I wonder for how long these pro-inflammatory effects of fibroblasts are observed? Previous studies in this direction demonstrated that such effects when focusing on immune cells, may hold for up to one week. Is this the case with fibroblasts?

This is an interesting point. We did analyze the expression of C3a in lung supernatant at the late metastatic stage and found that it is still significantly higher than controls (Figure 6e). We also showed that fibroblasts are the main source of complement factors in lungs (Figure 5d,e). Moreover, we analyzed human data (Fig. 5e) and in the revised manuscript we also analyzed C3 deposits in lung tissue sections from breast cancer patients (Figure 6f-h). These new analyses confirmed that stromal complement signaling is also operative at late stages. Thus, it seems that the effect is long lasting, relevant to human disease, and is evident also at late metastatic stage.

4. A very recent study demonstrated that following PTX therapy, the ECM is altered in the lungs, and as such this could also explain increased metastasis (Haj-Shomaly et al, Cancer Res 2022). This effect was not excluded from the analysis in this study. Especially when focusing on fibroblasts, are there any experiments which have tested ECM remodeling in response to chemotherapy?

To address this question, we assessed the changes in ECM at the lung metastatic niche by sirius red staining 24h following chemotherapeutic treatment. Interestingly, we found that doxorubicin significantly increased collagen deposition in the lungs as compared to controls, while cisplatin did not have a significant effect on collagen deposition in lungs in our model. These data are shown in new Supp. Fig. 5q-s. This is indeed very interesting and could be a topic of further study. The paper by Haj-Shomaly et al. was not published yet when we wrote our manuscript, and we are familiar with the beautiful work by the Shaked group. We now discussed this topic in the revised manuscript and cited this article. We thank the reviewer for this suggestion.

5. The authors claim that the CAFs are the main source of cells which contribute to the metastasis. However, some previous studies demonstrated that in addition to fibroblasts, immune cells and tumor cells (actually exosomes) can also promote chemotherapy-induced host responses promoting metastasis. I wonder whether the authors can clearly demonstrate that CAFs blockade or depletion change the metastatic effects in mice in response to chemotherapy.

We actually did not claim that CAFs are the main cell source that contributes to metastasis. We showed that CAFs are the main source of complement signaling following chemotherapy treatment, and this results in changes of the immune microenvironment. So, in that sense, immune cells are major players in forming a

hospitable metastatic niche and mediating therapy resistance. Indeed, our combination therapy targeted the complement receptors, C3aR and C5aR, expressed on recruited immune cells.

We agree with the reviewer that since CAFs are the main source of complement driving these changes, it would be interesting to target them specifically. Unfortunately, CAF blockade or depletion is not clinically applicable and there are no drugs that are specific enough to deplete only CAFs, especially given vast amounts of new data showing how diverse CAFs are (e.g. by single cell RNA-seq studies), and the fact that some CAF subpopulations were shown to have anti-tumorigenic effects (shown for example by Raghu Kalluri and colleagues in PDAC). Since we were seeking a mechanism that may be clinically translated to improve chemotherapy efficacy, we focused on blockade of signaling pathways, rather than depletion of CAFs.

6. Do the authors know whether the complement receptor antagonists do not affect the viability of cancer cells?

To address this concern, we incubated cancer cells with several concentrations of complement receptor antagonists and monitored their viability. We found that none of the concentrations tested were toxic for cancer cells. We included these results in the revised supplementary Fig. 9.

7. Fig. 4e, f showed lungs of doxorubicin-treated mice had increased infiltration of T cells expressing exhaustion/dysfunction markers such as TIM-3, TIGIT, and PD-1. This change in T cells markers might make the immunosuppressive TME niche in lungs more sensitive to immune checkpoint inhibitors such as in response to anti-TIGIT and anti-PD1 antibodies, something that may be worthy of future study of combination therapy involving immune checkpoint inhibitor with doxorubicin, and perhaps certain other drugs.

We agree with this excellent point. Indeed, such combinations would be an interesting and important point for future studies. Supportive evidence also comes from published studies such as Gao et al. *Front. Immunol.* 2020 PMID: 32194569, and the TONIC clinical trial (Voorwerk et al. *Nature Med.* 2019 PMID: 31086347). We added this interesting point to the revised discussion. We thank the reviewer for this suggestion.

8. This study suggested that the pro-tumorigenic adverse effects caused by the conventional MTD chemotherapy are largely due to the induction of tissue damage and tumor-promoting inflammation. To alleviate this kind of tissue damage, administering chemotherapy in a so-called “metronomic” low dose way could be a reasonable alternative to consider for future studies. Results of the ‘TONIC’ trial by Voorwerk et al (*Nat Med.* 2019 Jun;25(6):920-928) showed that 2 weeks of low-dose doxorubicin treatment on metastatic TNBC, inducing a more favorable immune-active TME with increased T cell infiltration and TCR diversity. Could the dose of doxorubicin used be a major factor causing the different effects on the TME? The dosing used in

current study was conventionally used MTD (5mg/kg) which led to the complement-induced immunosuppressive TME niche in lungs. Alizadeh et al (Cancer Res. 2014 Jan 1;74(1):104-18) used the same 4T1 cell line, but in a primary tumor setting. They demonstrated that MTD dosage of doxorubicin (5mg/kg) selectively eliminates MDSC in the spleen, blood, and tumor beds, which seems to be the opposite to what the current study is showing. Could it be that the primary tumors respond differently to doxorubicin therapy compared with the lung metastases in an adjuvant setting?

Indeed, metronomic treatment has different results than the MTD, as shown for example by the work of Kerbel et al. (Khan et al. npj Breast Cancer 2020 PMID: 32704531). We mentioned this point in our revised discussion. Regarding the reviewer's question on treatment in the presence of primary tumor vs. mets, there is no doubt that responses may be organ specific, as the microenvironments of each organ respond differentially. We actually also tested the effect of neo-adjuvant treatment in our system (before removal of primary tumor) and showed that it had a similar effect on recruitment of granulocytes to lungs, but no effect on their systemic levels (see graphs below).

9. Fig2C and Fig8C clearly show that regarding to prevention of lung metastasis from 4T1 tumor cell in an adjuvant setting, cisplatin alone is much more effective than the combination of doxorubicin and complement signaling inhibitor. If considering more toxicity might be the case with this combination treatment, a switch between doxorubicin and cisplatin should be a much better strategy, especially in a clinical setting – although as mentioned above, use of such drugs as monotherapy regimen is not normally done for adjuvant therapy of TNBC.

We agree with the reviewer that our results, as well as clinical data (e.g. from the PATTERN trial, mentioned by this reviewer) suggest that starting with cisplatin as a first line treatment may be more beneficial. However, the inclusion of platinum agents as neoadjuvant chemotherapy for TNBC remains clinically controversial and the use of platinum agents in the adjuvant setting is not recommended. Since the common clinical practice is treatment with doxorubicin, or a combination of AC, we set out to

study the adverse effects and suggest mechanism-based approaches to address them.

10. An obvious concern raised is the limitation of using just one tumor model. How about other mouse TNBC model such as EMT6? This study showed different drugs caused very different TME responses on the same model (4T1); logistically thinking, the same drug (doxorubicin) may have very different effects on different tumor models. I am not suggesting that an additional model be studied replicating all the experiments done in the 4T1 model – but something much more limited such as whether adriamycin or cisplatin have similar enhancing/suppressive effects on lung metastasis, respectively.

To address these suggestions, as mentioned above, we performed the same experiments with another model of TNBC, the EO771 model, in C57BL/6 mice. Strikingly, these additional set of experiments yielded similar results to our findings in the 4T1-based model (BALB/c background). The new results were integrated into multiple figures in the revised manuscript (Fig. 2f-h, Fig. 3d-f, Supp. Fig. 3, Supp. Fig 6e-g, Supp. Fig. 8d-f). Specifically, analysis of pulmonary metastatic load at late-stage showed that doxorubicin did not impact metastases formation as compared to control, while cisplatin treatment resulted in a trend towards lower metastatic load as compared to control ($p=0.07$) (New Fig. 2f-h). We also analyzed the lung immune microenvironment at an early metastatic stage and observed changes in lung infiltrating granulocytes and monocytes that were similar to what we found in 4T1 injected mice (Fig. 3d-f). Moreover, lungs of the doxorubicin-treated group had higher complement scores compared to control group (Supp. Fig. 8f). Most impressively, C3a protein levels were significantly elevated in lungs of doxorubicin-treated mice as compared to lungs derived from either cisplatin-treated mice or from control mice (Supp. Fig. 8d,e). Finally, we show that doxorubicin treatment led to increased accumulation of dysfunctional T cells in lungs of EO771-injected mice (Supp. Fig. 6e-g). We thank the reviewer for this suggestion.

11. Finally one final bit of information the authors should know concerns experiments in which tumor free normal mice are treated with a chemotherapy drug, and then tumor cells injected after such treatment, e.g. a day later, and finding that lung metastasis has increased. This is a very interesting result, however, similar experiments were first reported over 40 years ago and then a number of times since then. For example, see Carmel RJ et al, *Cancer Res* 1977 (“The effect of cyclophosphamide and other drugs in the incidence of pulmonary metastases in mice”) and Yamauchi K et al, *Cancer Res* 2008 (“Induction of cancer metastasis by cyclophosphamide pre-treatment of host mice: an opposite effect of chemotherapy”). I think it should also be noted that the literature reporting chemotherapy induction or promotion of metastasis is almost entirely preclinical in nature, and an obvious question is whether there is any definitive clinical trial evidence for chemotherapy promotion of metastasis in patients. By the

way the two papers mentioned above both showed that pretreatment of normal mice with cyclophosphamide was a powerful inducer of metastasis. So is this inconsistent with the use of the 'AC' regimen in clinical adjuvant therapy of TNBC?

We thank the reviewer for providing us with this interesting information. Obviously, experiments where healthy individuals are treated with chemotherapy and then the effect on metastatic spread is analyzed are not ethically feasible in humans, explaining why all such data is pre-clinical. These experiments, reported in Figure 1 of our manuscript were performed to test our hypothesis that tissue damage may activate fibroblasts and affect the pre-metastatic niche and they merely set the stage for the more clinically relevant experiments performed in an adjuvant setting.

As for the reviewer's question regarding the consistency of our findings with the AC regimen, we performed (also mentioned above) experiments where mice were treated with DOX+CTX following resection of their primary tumor and found similar results to treatment with doxorubicin alone: AC treatment resulted in a significant increase in recruitment of granulocytes, monocytes and T cells. Moreover, while the combination treatment (DOX+CTX) attenuated the progression rate, the metastatic burden at end-stage was not different than dox treatment or no treatment. These results are presented in new supp. Fig. 5. Thus, the combination treatment, commonly used in TNBC patients had similar adverse effects. While we are not clinicians, we do hope that our findings may instruct the design of better therapeutic strategies.

We would like to thank the reviewer for being thorough and positive and for their interesting and thought-provoking comments.

REVIEWERS' COMMENTS

Reviewer #1 (Remarks to the Author):

The authors addressed my major concerns.

Reviewer #2 (Remarks to the Author):

I was reviewer 2 for this manuscript originally. In this revised version of their manuscript, Monteran and colleagues have presented a substantial amount of new data - most importantly to validate their key findings in another syngeneic model of a different strain. These have clearly explained how they have addressed the comments raised and clearly marked out the changes in the revised manuscript.

My only remaining comments are minor but should be addressed to help others navigate the latter part of this manuscript

1: Fig. 8e,f. The authors explain that the a values in 8e relate to the slope values presented in 8f. However, this doesn't quite make sense as for the DOX+SB group the a value is 0.042 where as the red line in 8f represents the mean slope value. But the red line is not at 0.042 - looks more like 0.1

2: Fig. 7, Fig. 8 and Supplementary Fig. 9. I requested the authors clarify how the data in SFig. 9 relates to Fig. 8 and whether Fig. 8 is a continuation of the animal experiment presented in Fig. 7. The authors now state that Supplementary Fig. 9 relates to Fig. 7 and Fig. 8 but that is not very helpful. The authors need to state within the panels for each of these these data relate to each other. For example. What is the relationship between Fig. 8b-d with the data shown in Supplementary Fig. 9k-l. Are these the same experiment? The y axes labelled number of mets/mouse CT. What does CT stand for? Why do some panels (S8k) show 0 scores but others (8c) only have values >0? Why do two of the columns in 8c have a single 0 value. I just cannot get my head around this. The authors have plenty of space in Fig. 8 - why do they not just show all of the data in one place?

All of this is not helped by the Fig. 8 legend. The authors don't even start by telling us what tumor cells were injected - I assume 4T1. Same in Fig. 7 legend - again what cells were injected.

Reviewer #3 (Remarks to the Author):

I have completed review of this revised manuscript, being the 3rd referee for the original submission. On balance, I think the authors have done a very commendable job in thoroughly revising the manuscript, and clarifying certain issues or concerns I had in their rebuttal. I have not looked over in detail the responses to the two other reviews, but they also seem to be extremely detailed and thorough. I think that this manuscript is now acceptable for publication and I only have a few general comments that the authors might consider for comment and minor revision. I think one of the difficult issues in similar studies which claim to show a pro-metastatic effect of commonly used anti-cancer drugs (or relatively new ones as well) is the issue of clinical relevance. What I mean by that is that if it was generally true that drugs such as adriamycin, or the combination of adriamycin plus cisplatin (AC), promote tumor progression rather than the opposite, why would any patient with breast cancer, for example, want to be treated with these drugs?! The authors have actually addressed this question and conundrum in their rebuttal response by pointing out that there are likely both desirable, and possibly undesirable, effects in terms of overall impact on tumor growth, and that it is the balance, i.e., net outcome, of the two opposing effects that determine ultimate clinical benefit, or lack thereof. I agree with this argument and as such, I think it is important to highlight this point in the manuscript. In this regard, I think a suitable place to do so would be at the end of the Discussion, which is where the editors of many journals ask authors to comment on potential weaknesses or concerns of an overall study. Putting forward the 'balance' (net outcome) argument would be a way of dealing with this point.

Another point that might be worth considering in terms of a minor revision is that the authors undertook experiments where they evaluated the AC combination and found that the pro-metastatic effect of adriamycin alone is still observed, despite cisplatin on its own having an anti-metastatic effect, was a bit disappointing! I had speculated that if cisplatin reverses the pro-metastatic effect of adriamycin, this would have been a seminal observation in the field of cancer chemotherapy and tumor progression. But it is what it is. In this regard, I would only add one other point or caveat for the authors to consider in future studies – not for this paper. In certain areas such as N. America, the AC regimen is often followed by a weekly paclitaxel regimen (“AC→T”), as previously mentioned. Therefore it still might be possible that duplicating this regimen in mice might show that the pro-metastatic effect of AC is prevented. Again, the authors should consider just mentioning this possibility, and that perhaps it will be evaluated in future studies.

In summary, with these relatively minor revisions (though they are important in my opinion) I would recommend acceptance of this manuscript and in doing so, commend the authors for the revisions that have been made, and for the clarifications they have summarized in their rebuttal letter. This is an important study, which as I previously mentioned, links several different and diverse aspects of cell and cancer biology and has a number of potentially important biological and clinical ramifications.

Second round:

Reviewer #1 (Remarks to the Author):

The authors addressed my major concerns.

Reviewer #2 (Remarks to the Author)

I was reviewer 2 for this manuscript originally. In this revised version of their manuscript, Monteran and colleagues have presented a substantial amount of new data - most importantly to validate their key findings in another syngeneic model of a different strain. These have clearly explained how they have addressed the comments raised and clearly marked out the changes in the revised manuscript.

We thank the reviewer for their kind words.

My only remaining comments are minor but should be addressed to help others navigate the latter part of this manuscript

1: Fig. 8e,f. The authors explain that the a values in 8e relate to the slope values presented in 8f. However, this doesn't quite make sense as for the DOX+SB group the a value is 0.042 where as the red line in 8f represents the mean slope value. But the red line is not at 0.042 - looks more like 0.1

The reviewer is correct. Thank you for pointing this out. Indeed, the red lines represent the median values of the slopes. We corrected the figure legend accordingly. We

apologize for this mistake.

2: Fig. 7, Fig. 8 and Supplementary Fig. 9. I requested the authors clarify how the data in SFig. 9 relates to Fig. 8 and whether Fig. 8 is a continuation of the animal experiment presented in Fig. 7. The authors now state that Supplementary Fig. 9 relates to Fig. 7 and Fig. 8 but that is not very helpful. The authors need to state within the panels for each of these these data relate to each other.

For example. What is the relationship between Fig. 8b-d with the data shown in Supplementary Fig. 9k-l. Are these the same experiment?

Yes, this is the same experiment (performed multiple times). We better clarified this in the revised manuscript (highlighted in yellow).

The y axes labelled number of mets/mouse CT. What does CT stand for?

"CT" refers to CT scans. We changed the Y axes labels accordingly.

Why do some panels (S8k) show 0 scores but others (8c) only have values >0? Why do two of the columns in 8c have a single 0 value. I just cannot get my head around this. The authors have plenty of space in Fig. 8 - why do they not just show all of the data in one place?

The panel in Sup 9k-l present the data from all the mice, some of which did not develop metastases, and hence they have 0 values. As explained in the text, the analysis in main Fig. 8c,d was performed only of mice with metastases, and hence no 0 values should be there. We double checked and the graphs are now correct. We thank the reviewer for pointing this out.

All of this is not helped by the Fig. 8 legend. The authors don't even start by telling us what tumor cells were injected - I assume 4T1. Same in Fig. 7 legend - again what cells were injected.

We apologize that this was not sufficiently clear. We now specifically stated in each figure legends which cells that were injected.

Reviewer #3 (Remarks to the Author)

I have completed review of this revised manuscript, being the 3rd referee for the original submission. On balance, I think the authors have done a very commendable job in thoroughly revising the manuscript, and clarifying certain issues or concerns I had in their rebuttal. I have not looked over in detail the responses to the two other reviews, but they also seem to be extremely detailed and thorough. I think that this manuscript is now acceptable for publication and I only have a few general comments that the authors might consider for comment and minor revision. I think one of the difficult issues in similar studies which claim to show a pro-metastatic effect of commonly used anti-cancer drugs (or relatively new ones as well) is the issue of clinical relevance. What I mean by that is that if it was generally true that drugs such as adriamycin, or the combination of adriamycin plus cisplatin (AC), promote tumor progression rather than the opposite, why would any patient with breast cancer, for example, want to be treated with these drugs?! The authors have actually addressed this question and conundrum in their rebuttal response by pointing out that there are likely both desirable, and possibly undesirable, effects in terms of overall impact on tumor growth, and that it is the balance, i.e., net outcome, of the two opposing effects that determine ultimate clinical benefit, or lack thereof. I agree with this argument and as such, I think it is important to highlight this point in the manuscript. In this regard, I think a suitable place to do so would be at the end of the Discussion, which is where the editors of many journals ask authors to comment on potential weaknesses or concerns of an overall study. Putting forward the 'balance' (net outcome) argument would be a way of dealing with this point.

As suggested, we added a sentence at the end of the discussion section, referring to the balance between the beneficial and adverse effect of chemotherapy.

Another point that might be worth considering in terms of a minor revision is that the authors undertook experiments where they evaluated the AC combination and found that the pro-metastatic effect of adriamycin alone is still observed, despite cisplatin on its own having an anti-metastatic effect, was a bit disappointing! I had speculated that if cisplatin reverses the pro-metastatic effect of adriamycin, this would have been a seminal observation in the field of cancer chemotherapy and tumor progression. But it is what it is. In this regard, I would only add one other point or caveat for the authors to consider in future studies – not for this paper. In certain areas such as N. America, the AC regimen is often followed by a weekly paclitaxel regimen ("AC→T"), as previously mentioned. Therefore it still might be possible that duplicating this regimen in mice might show that the pro-metastatic effect of AC is prevented. Again, the authors should consider just mentioning this possibility, and that perhaps it will be evaluated in future studies.

In summary, with these relatively minor revisions (though they are important in my opinion) I would recommend acceptance of this manuscript and in doing so, commend the authors for the revisions that have been made, and for the clarifications they have summarized in their rebuttal letter. This is an important study, which as I previously mentioned, links several different and diverse aspects of cell and cancer biology and has a number of potentially important biological and clinical ramifications.